# EVOLUTIONARY PROFILES FOR PROTEIN FITNESS PREDICTION

## ABSTRACT

Predicting the fitness impact of mutations is central to protein engineering but constrained by limited assays relative to the size of sequence space. Protein language models (pLMs) trained with masked language modeling (MLM) exhibit strong zero-shot fitness prediction; we provide a unifying view by interpreting natural evolution as implicit reward maximization and MLM as inverse reinforcement learning (IRL), in which extant sequences act as expert demonstrations and pLM log-odds serve as fitness estimates. Building on this perspective, we introduce EvoIF, a lightweight model that integrates two complementary sources of evolutionary signal: (i) within-family profiles from retrieved homologs and (ii) cross-family structural–evolutionary constraints distilled from inverse folding logits. EvoIF fuses sequence–structure representations with these profiles via a compact transition block, yielding calibrated probabilities for log-odds scoring. On ProteinGym (217 mutational assays; >2.5M mutants), **EvoIF and its MSA-enabled variant** achieve state-of-the-art or competitive performance **with significantly reduced training data** (0.15% compared to large-scale pLMs). Ablations confirm that within-family and cross-family profiles are complementary, improving robustness across function types, MSA depths, taxa, and mutation depths.

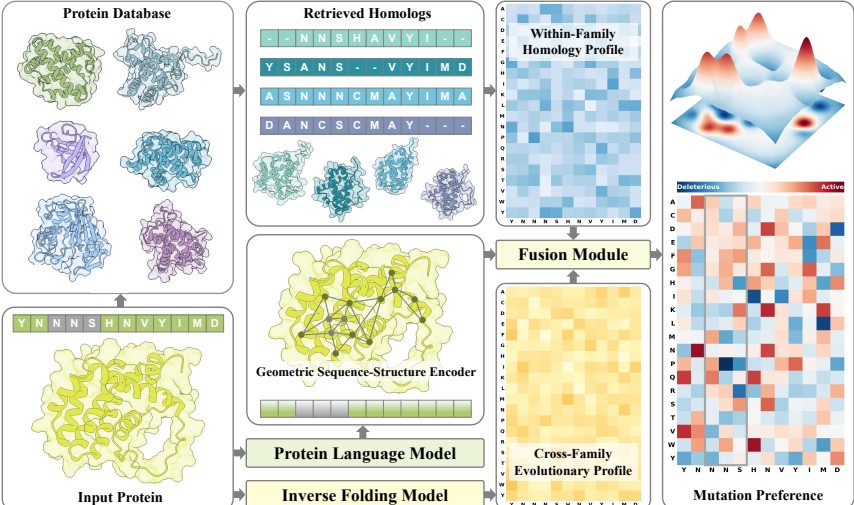

Figure 1: Overview of EvoIF.

## 1 INTRODUCTION

Protein evolution is driven by selective pressure: mutations that preserve or enhance function are preferentially retained, whereas deleterious ones are eliminated [10]. The success of a protein variant within this evolutionary landscape is quantified by its fitness, a measure of its functional viability and contribution to an organism's survival. Mapping this sequence–function relationship, commonly referred to as the fitness landscape, is therefore a central challenge in molecular biology. Accurate prediction of mutational fitness forms the foundation of rational protein design [36, 31], enabling the engineering of enzymes with enhanced catalytic efficiency, antibodies with improved affinity,

and biologics with increased stability, thereby addressing critical problems in therapeutics, materials science, and sustainability.

Protein fitness prediction is constrained by the scarcity of experimental measurements relative to the vastness of protein space [1]. Consequently, self-supervised methods for protein representation learning have become essential for protein fitness prediction [12, 25, 50]. Recently, protein language models (pLMs) including ESM series [35, 19] and their structure-informed variants [14], trained through masked language modeling (MLM), have demonstrated remarkable zero-shot capabilities in protein fitness prediction [30]. These models can predict the impact of mutations on protein function without additional training specific to particular protein families, sometimes achieving performance comparable to specially trained models. Current state-of-the-art approaches, including AIDO-Protein-RAG [18] and VenusREM [42], further boost performance by integrating homologous sequences as evolutionary context.

Although the encouraging results mentioned above, current methods still confronted with several substantial challenges:

**Issue 1. Most protein language models are trained using the MLM task , yet there is still a lack of a reasonable explanation** for why MLM can serve as a proxy task for protein fitness prediction.

**Issue 2. Current approaches tend to focus heavily on scaling model parameters and training data,** yet the performance gain in protein fitness prediction remain marginal (Figure 3). Moreover, the computational requirements for pre-training and further fine-tuning such large-scale models can be extremely high, which may restrict their practical applicability in resource-constrained settings.

**Issue 3. Existing models have not fully considered the comprehensive modeling of protein evolutionary information.** For sequence evolution information, researchers have applied Multiple Sequence Alignment (MSA) [4] for modeling. In contrast, Inverse Folding (IF) [13] has been developed to model cross-family structural evolutionary information. Notably, MSA relies solely on sequences, while IF depends solely on structure. Therefore, for a protein with both sequence and structure, it is natural to construct a comprehensive evolutionary model that incorporates both its sequence and structural information. However, this aspect remains underexplored. The majority of research treats structure merely as part of protein representation, overlooking the evolutionary information embedded within it.

To address the issues mentioned above, this paper makes the following contributions:

1) We first propose that protein evolution can be viewed as an implicit reward-maximization process in which natural selection acts as an expert that iteratively selects high-fitness sequences; the resulting extant sequences therefore constitute an expert demonstration set. From this perspective, MLM pre-training aligns with inverse reinforcement learning (IRL) [26]: recover the latent reward (fitness) from the observed expert's behaviors (protein sequences). We show that the maximum-likelihood objective of MLM coincides with the maximum-entropy IRL loss [52]; accordingly, the log-odds ratio produced by a pLM provides an estimate of protein fitness.

2) We explicitly incorporate sequence evolutionary information from homologous sequences of the same family into the model. This information is obtained through sequence similarity searches [4], or structure similarity searches such as Foldseek [44], to identify the most closely related sequences within the same family. These sequences exhibit the most direct sequence or structure homology and have been shown to be beneficial for predicting protein fitness [18, 42]. This approach can be viewed as a form of in-context reinforcement learning, where homologous sequences act as supplementary expert demonstrations. By providing family-specific contextual information, these homologous sequences enhance the basis for protein fitness prediction.

3) Furthermore, we attempt to explicitly integrate cross-family structural evolutionary information into the model. While there has been extensive research on modeling sequence MSA, it is ultimately the three-dimensional structure encoded by these sequences that determines protein function and activity. During protein evolution, accumulated mutations lead to corresponding structural changes, thereby driving fitness evolution [37]. The IF model can predict high-confidence amino acid sequences compatible with a given backbone structure, effectively performing the inverse task of structure prediction. Since it is trained on natural protein structures and sequences, it is capable of capturing the complex distribution patterns of protein sequences shaped by evolutionary dynamics. Recent studies [37, 5] suggest that the IF model tends to select amino acids similar to natural variants, indicating

that it has internalized key structural–evolutionary couplings across families. Therefore, we treat the likelihood values provided by the IF model as a compact structural evolutionary profile and explicitly incorporate it into the model to provide cross-family evolutionary information.

In summary, we propose **EvoIF**, a lightweight network that combines (i) within-family evolutionary information from homologous sequence MSA retrieved through sequence or structure searches, and (ii) cross-family evolutionary information embedded in the IF likelihood values, together with its MSA-enabled variant, **EvoIF-MSA**. By effectively integrating evolutionary features from homologous sequences and cross-family structures, EvoIF offers a data-efficient solution: in the deep mutational scanning (DMS) [6] experiment of over 2.5 million mutants across 217 proteins in ProteinGym [30], its performance is state-of-the-art or comparable, while **requiring only 0.15% of the training data** used by large-scale pLMs. Additional ablation studies demonstrate that these different dimensions of evolutionary information complement each other well and show strong robustness as training data is further reduced. Together, these results suggest that EvoIF is an efficient and robust network for modeling evolutionary information. EvoIF provides accurate protein evolutionary profiles, and due to its lightweight nature, it enables fine-tuning for specific proteins or tasks, offering broad benefits.

## 2 METHOD

We present EvoIF, a data-efficient framework for protein fitness prediction that (i) encodes sequence–structure context with a lightweight sequence–structure backbone (Section 2.3) and (ii) injects evolutionary information through two compact profiles: a structure-retrieved homology profile and an inverse folding profile (Section 2.4). The fused probabilities enable zero-shot log-odds scoring (Section 2.1) consistent with the IRL view (Section 2.2).

### 2.1 PROTEIN LANGUAGE MODELS FOR FITNESS PREDICTION

**Definition.** The protein fitness landscape describes how a protein's function changes with its sequence, which can be quantitatively measured by methods like DMS [6]. In DMS, **fitness** is a quantitative measure of a protein variant's functional performance under specific selective pressure. Fitness $F$ is calculated as the relative change in a variant's abundance $N^{\text{mt}}$ from the pre-selection to the post-selection population, normalized to the change in the wild-type's abundance $N^{\text{wt}}$:

$$F(S^{\text{mt}}, S^{\text{wt}}) = \log\left(\frac{N_{\text{post}}^{\text{mt}}/N_{\text{pre}}^{\text{mt}}}{N_{\text{post}}^{\text{wt}}/N_{\text{pre}}^{\text{wt}}}\right) \tag{1}$$

where a positive fitness value indicates a beneficial mutation, a negative value indicates a deleterious mutation, and a value near zero suggests a neutral effect on the protein's function. The specific biological meaning of fitness score depends directly on the type of selective pressure applied.

**Notation and assumption.** We focus on substitutions and, consistent with common practice, assume that a small number of substitutions do not alter the protein's backbone structure [43, 40, 50, 17, 42, 41, 18]. Given a wild-type protein with sequence $S^{\text{wt}}$ and structure $X^{\text{wt}}$, its mutant has a sequence $S^{\text{mt}}$ that differs from $S^{\text{wt}}$ at the mutation sites, while its backbone structure remains unchanged ($X^{\text{wt}} = X^{\text{mt}}$). The objective is to develop an unsupervised model that predicts the fitness score for each mutant, quantifying its functional change relative to the wild-type.

**Common practice.** pLMs are trained on the MLM objective, learning to predict residues at masked positions based on the surrounding context [19, 46]. As detailed in Meier *et al.* [24], this capability allows pLMs to score sequence variations by calculating the log-odds ratio between the mutant and wild-type proteins for a set of mutations $\mathcal{M}$:

$$\mathcal{L}_{\text{MLM}} = -\sum_{i \in \mathcal{M}} \log P(s_i \mid S_{\backslash \mathcal{M}}) \tag{2}$$

$$\hat{F}(S^{\text{mt}}, S^{\text{wt}}) = \sum_{i \in \mathcal{M}} \log P\left(s_i = s_i^{\text{mt}} \mid S_{\backslash \mathcal{M}}\right) - \log P\left(s_i = s_i^{\text{wt}} \mid S_{\backslash \mathcal{M}}\right) \tag{3}$$

Here, $S_{\backslash \mathcal{M}}$ denotes the input sequence with each mutated position in $\mathcal{M}$ masked. This scoring method assumes an additive model for multiple mutation sites. In the zero-shot setting, the model evaluates the sequence using a single forward pass.

## 2.2 PROTEIN EVOLUTION AS A MARKOV DECISION PROCESS

We formalize protein evolution as a Markov decision process (MDP) where the **state space** $\mathcal{S}$ consists of all possible protein sequences, the **action space** $\mathcal{A}$ represents point mutations acting on amino acid residues (with deterministic transition dynamics), the **reward function** $R : \mathcal{S} \to \mathbb{R}$ encodes selective pressure (not known *a priori*), and **expert demonstrations** $\mathcal{D}$ contain observed evolutionary trajectories of stable proteins under natural selection.

This MDP formulation enables the application of IRL to protein evolution. We explicitly adopt three simplifying assumptions: (1) **Markovian property**: Transition probabilities depend solely on the current sequence state, neglecting epistatic dependencies on historical mutations [39]. (2) **Stationary reward**: Fitness landscapes are assumed time-invariant, though environmental shifts may alter selection pressures. (3) **Expert optimality**: Observed sequences are treated as optimal with respect to $R$, despite evolutionary constraints such as local optima, since the evolutionary traversed space may be limited compared to the vast protein sequence space.

Although based on simplifying assumptions, the MDP abstraction captures core dynamics of protein evolution. Crucially, it allows us to interpret natural selection as an expert policy $\pi^*$ that maximizes long-term fitness. Unlike standard reinforcement learning (RL), which finds an optimal policy to maximize rewards, IRL [26] works backward, inferring the reward function that best explains expert trajectories. Specifically, Maximum Entropy IRL (MaxEnt IRL) [52] refines this by assuming expert actions follow a Boltzmann distribution proportional to expected reward.

The MLM training objective of pLMs aims to maximize the log-likelihood of sequences by learning to predict masked amino acids given their context (Equation 2). Maximum Entropy IRL, in turn, models the probability of an expert trajectory $\zeta$ under a reward function $R_\theta$ as

$$P_\theta(\zeta) = \frac{\exp\left(R_\theta(\zeta)\right)}{Z_\theta}, \; Z_\theta = \sum_{\zeta'} \exp\left(R_\theta(\zeta')\right) \tag{4}$$

Here, $Z_\theta$ is the partition function that normalizes probabilities across all possible trajectories $\zeta'$. Given a dataset of expert demonstrations $\mathcal{D}$, the MaxEnt IRL log-likelihood is

$$\mathcal{L}_{\text{IRL}}(\theta) = \frac{1}{|\mathcal{D}|} \sum_{\zeta \in \mathcal{D}} \log P_\theta(\zeta) = \frac{1}{|\mathcal{D}|} \sum_{\zeta \in \mathcal{D}} R_\theta(\zeta) - \log Z_\theta \tag{5}$$

So maximizing $\mathcal{L}_{\text{IRL}}$ selects the reward best explaining the trajectories and is equivalent to minimizing the MLM objective (Equation 2). Under the MaxEnt–Boltzmann assumption (Equation 4), $P_\theta(S) \propto \exp\left(R_\theta(S)\right)$, so the pLM's log-probabilities provide an affine surrogate for the reward. Consequently, reward differences are proportional to log-probability differences; in particular

$$\Delta R_\theta\left(S^{\text{mt}}, S^{\text{wt}}\right) = \sum_{i \in \mathcal{M}} \left[ \log P_\theta\left(s_i^{\text{mt}} \mid S_{\setminus \mathcal{M}}\right) - \log P_\theta\left(s_i^{\text{wt}} \mid S_{\setminus \mathcal{M}}\right) \right] \tag{6}$$

Under this assumption, pLM log-probabilities estimate the reward (up to an affine transformation). Viewing experimental fitness as a relative reward, Equation 3 then admits a principled interpretation: pLM log-odds estimate the reward difference between mutant and wild-type, serving as a zero-shot predictor for fitness $F(S^{\text{mt}}, S^{\text{wt}})$.

A common practice in protein fitness prediction is to supplement pLMs with evolutionary information from homologous sequences, which has been shown to further boost performance [18, 42]. Similarly, in large language models, a technique called *self-evolution* has emerged, where models use prior problem-solving trajectories as *context* to improve their reasoning and agentic abilities [8, 11, 47, 51]. This parallel suggests an intuitive explanation: just as humans learn from examples and adapt their reasoning based on relevant context, both protein language models and general language models can benefit from incorporating evolutionary trajectories as contextual demonstrations. In the protein domain, homologous sequences retrieved via sequence similarity searches [4] or structure-based searches [44] provide evolutionary trajectories that act as expert demonstrations, constraining the solution space to biologically plausible mutations.

## 2.3 SEQUENCE–STRUCTURE MODEL FOR FITNESS PREDICTION

While pLMs are powerful for predicting mutational effects, incorporating 3D structural information has emerged as a common strategy to enhance their predictive performance [50, 40, 42]. Our model

builds upon S2F in Zhang *et al.* [50] to enhance mutational effect prediction. We augment pLM features with geometric context by using a graph neural network (GNN) to process protein backbone structure. Specifically, we use Geometric Vector Perceptron (GVP) [15] networks for message passing on a protein's graph representation. The GVP module ensures SE(3)-invariance for scalar features and SE(3)-equivariance for vector features, which is crucial for handling 3D structural data. This architectural choice follows prior evaluations demonstrating GVP's effectiveness for fitness prediction [50], which we further validate through ablation studies comparing GVP with alternative architectures such as GearNet (see Section E.6), confirming GVP's superior performance across all metrics.

Formally, the hidden state of residue $i$ at layer $l$, $\boldsymbol{h}_i^{(l)}$, is represented by $d$-dim scalar features and $d'$-dim vector features. Initial node features are set using ESM-2 embeddings, with $\boldsymbol{h}_i^{(0)} = \left(\text{ESM-2}\left(s_i \mid \boldsymbol{S}_{\backslash\mathcal{M}}\right), \mathbf{0}\right)$. Edge features $\boldsymbol{e}_{(j,i)}$ encode pairwise distances and coordinate differences using Radial Basis Function (RBF) kernels. Message passing is performed using GVP modules, which process both scalar and vector features while ensuring SE(3)-invariance and SE(3)-equivariance, respectively. Each GVP layer is followed by a feed-forward network:

$$\begin{aligned} \boldsymbol{h}_i^{(l+0.5)} &= \boldsymbol{h}_i^{(l)} + \frac{1}{|\mathcal{N}(i)|} \sum_{j \in \mathcal{N}(i)} \text{GVP}\left(\boldsymbol{h}_j^{(l)}, \boldsymbol{e}_{(j,i)}\right) \\ \boldsymbol{h}_i^{(l+1)} &= \boldsymbol{h}_i^{(l+0.5)} + \text{GVP}\left(\boldsymbol{h}_i^{(l+0.5)}\right) \end{aligned} \tag{7}$$

Finally, the scalar features from the last layer, $\boldsymbol{h}_i^{(L)}$, are used to predict the residue type via a linear layer.

### 2.4 Evolutionary Profiles for Fitness Prediction

**Sequence and structure profiles.** MSA [4] serve as a fundamental tool in computational protein modeling, capturing evolutionary relationships and co-evolutionary signals. While MSA-based approaches are widely applied to diverse tasks like protein structure prediction, function prediction, and design, and remain a mainstream strategy for protein fitness prediction, the raw MSA format poses practical challenges. Its variable length and depth, as well as potential alignment errors, may compromise both accuracy and efficiency in scaled models. As a result, recent research in protein design [9], structure prediction [32], and optimization [23] has converged on using evolutionary profiles as a more compact and manageable evolutionary representation. For a protein with $n$ aligned sequences $\{S_1, S_2, \ldots, S_n\}$, each of length $L$, the evolutionary profile is represented as a matrix $\boldsymbol{P} \in \mathbb{R}^{L \times 21}$, where each entry $\boldsymbol{P}_{ij}$ denotes the frequency of amino acid $A_j$ (including one special gap character "-") at position $i$ across the aligned sequences:

$$\boldsymbol{P}_{ij} = \frac{1}{n} \sum_{k=1}^{n} \mathbb{I}\left(S_{k,i} = A_j\right) \tag{8}$$

Here, $\mathbb{I}(\cdot)$ is the indicator function, $A_j \in A \cup \{-\}$ and $A$ denotes the set of 20 standard amino acids. In addition to using **sequence profiles**, Tan *et al.* [42] also constructs evolutionary profiles from structurally within-family homologous sequences via Foldseek [44]. Such **structure profiles** broaden the scope of this compact representation beyond pure within-family sequence-based homology.

**Inverse folding profile.** While evolutionary profiles are a powerful and compact representation of evolutionary information, their quality is directly dependent on the homologous search used to construct them. This process suffers from two primary limitations: (1) **Limited scope**: the search often retrieves only the most closely related homologs, lacking coverage of the broader cross-family structural evolutionary landscape; (2) **Computational cost**: searching massive databases for homologs is computationally expensive and time-consuming, often taking tens of minutes for a single protein. Given these limitations, we explore how to integrate evolutionary information more efficiently and comprehensively, and attempt to capture broader cross-family evolutionary profiles. Recent work [37, 5] shows that inverse-folding models trained on structure-conditioned sequence recovery tend to favor amino acid choices that mirror natural variation. Because they are trained on natural protein structures and sequences, they can capture the complex distribution patterns of protein sequences shaped by evolutionary dynamics. We therefore take the likelihood provided by inverse-folding models as an informative evolutionary profile.

**Fusion module.** To effectively integrate the complementary information from sequence–structure modeling and evolutionary profiles, we design a fusion strategy that processes each probability distribution through a transformer layer as transition block before combination. Given the S2F structural representation probabilities $\boldsymbol{P}^{\text{S2F}} \in \mathbb{R}^{L \times 21}$, within-family structural homologs' profile probabilities $\boldsymbol{P}^{\text{struct}} \in \mathbb{R}^{L \times 21}$, and cross-family inverse folding profile probabilities $\boldsymbol{P}^{\text{IF}} \in \mathbb{R}^{L \times 21}$, where $L$ is the sequence length, the model's predicted logits is obtained by:

$$\boldsymbol{P}_{\text{final}} = \text{softmax}(\boldsymbol{P}^{\text{S2F}} + \text{Transition}(\boldsymbol{P}^{\text{struct}}) + \text{Transition}(\boldsymbol{P}^{\text{IF}})) \tag{9}$$

This fusion strategy allows the model to capture contextual relationships within each probability distribution through the transition block, then combine the processed distributions through addition and normalize the result to ensure valid probability distributions.

## 2.5 PRE-TRAINING AND INFERENCE

We adopt the pre-training and inference recipe outlined in Devlin *et al.* [3] and Zhang *et al.* [50]. For pre-training, we employ the MLM objective on the non-redundant subset of the CATH v4.3.0 dataset [38], comprising 30,948 experimental protein structures. Instantiating the standard MLM loss (Equation 2) with the fused probabilities in Equation 9, we obtain the loss function:

$$\mathcal{L}_{\text{MLM}}^{\text{fusion}} = - \sum_{i \in \mathcal{M}} \log P_{\text{final}}(s_i \mid S_{\backslash \mathcal{M}}) \tag{10}$$

where $\mathcal{M}$ represents the set of masked positions, $s_i$ is the true amino acid at position $i$ from the training sequence, and $P_{\text{final}}$ is obtained from the multi-source fusion in Equation 9. The weights of the ESM-2 and ProteinMPNN models are frozen, with only the profile transition blocks for the external profiles and the GVP layers for the structure graphs remaining trainable. Comprehensive training details are provided in Appendix D.1.

During inference, fitness prediction follows the log-odds approach outlined in Equation 3, where the model calculates the log-odds ratio between mutant and wild-type sequences to estimate the functional impact of mutations. Specifically, for a mutant sequence $S^{\text{mt}}$ and wild-type sequence $S^{\text{wt}}$ with mutation sites $\mathcal{M}$, the predicted fitness is computed as:

$$\hat{F}(S^{\text{mt}}, S^{\text{wt}}) = \sum_{i \in \mathcal{M}} \left[ \log P_{\text{final}} \left( s_i = s_i^{\text{mt}} \mid S_{\backslash \mathcal{M}} \right) - \log P_{\text{final}} \left( s_i = s_i^{\text{wt}} \mid S_{\backslash \mathcal{M}} \right) \right] \tag{11}$$

where $P_{\text{final}}$ is the fused probability distribution from Equation 9, and $S_{\backslash \mathcal{M}}$ denotes the input sequence with each mutated position in $\mathcal{M}$ masked.

We refer to this pre-training and inference setup as our **base model**, **EvoIF** (MSA-free). To enable fair comparisons with alignment-dependent baselines, we also report an **MSA-enabled** variant, **EvoIF-MSA**, following Zhang *et al.* [50]. At inference time, EvoIF is ensembled with the MSA-only method GEMME [16] by summing standardized $z$-scores. This post hoc procedure does not modify the EvoIF architecture or its training protocol and is applied only when an MSA is available.

## 3 EXPERIMENTS

### 3.1 EXPERIMENTAL SETTINGS

**Dataset.** ProteinGym [30] is a widely-used benchmark for protein mutation effect prediction. It contains 217 DMS assays with over 2.5 million substitution mutations, covering key functional properties like stability, binding, and activity. The curated experimental DMS data provide standardized sequences, predicted structures, and evolutionary information for fair model comparison.

**Evaluation metrics.** We employ five standard metrics: Spearman correlation, AUC, MCC, NDCG, and top-10% recall. All metrics are computed using standardized scripts from the ProteinGym repository. Detailed descriptions of all metrics are provided in Appendix D.4. Fitness values in ProteinGym are normalized as a preprocessing step, specifically centered and normalized to the interval $[0, 1]$. Additionally, fitness is inherently a normalized metric. Since our evaluation primarily uses Spearman correlation, which measures ranking accuracy, normalization does not affect our results, as rank-order relationships are invariant under monotonic transformations.

**Comparison methods.** We benchmark against a broad set of state-of-the-art unsupervised methods, categorized as follows; detailed descriptions of all methods are provided in Appendix C.1:

- **Sequence-based models**: ProGen2 XL [27], CARP-640M [49], ESM-2-650M [19].

- **Alignment-dependent models**: DeepSequence [7], MSA Transformer [33], Tranception L with retrieval [28], EVE [7], GEMME [16], TranceptEVE L [29].

- **Inverse folding models**: ProteinMPNN [2], MIF [48], ESM-IF [14].

- **Sequence–structure hybrid models**: MIF-ST [48], ProtSSN [43], SaProt [40], S2F [50], S3F [50], ProtSST ($K$=2048) [17].

- **Structure- and MSA-hybrid models**: S2F-MSA [50], S3F-MSA [50], VenusREM [42], AIDO-Protein-RAG 16B [41, 18].

## 3.2 MAIN RESULTS

Table 1 shows the results of our method and comparison methods. We observe that our method achieves superior or comparable performance across a wide range of baselines in different settings. EvoIF significantly outperforms sequence-based pLMs, MSA-based approaches, and inverse folding models. This indicates that sequence- or structure- evolutionary signals alone are insufficient to reflect the actual evolutionary fitness landscape. Compared with hybrid models that integrate both sequence and structural features, EvoIF also achieves the best performance, surpassing previous S2F and S3F variants. The only exception is ProtSST, which relies on more than 600 times the training data together with a highly complex substructure clustering process and extensive hyperparameter tuning. When further combined with MSA signals, our method establishes a new state-of-the-art, outperforming or comparable to the previously best sequence–structure hybrid models and structure–MSA hybrid models. It further demonstrates remarkable computational efficiency, with training over $10^9$ times faster than AIDO Protein-RAG-16B and over 900 times faster than VenusREM (Figure 3).

**These results highlight both the effectiveness and efficiency of EvoIF and EvoIF-MSA.** Our method enables much shorter training times than existing large-scale baselines and demonstrate strong capability in capturing evolutionary information.

Table 1: **Overall results on ProteinGym benchmark. Bold** and underline indicate the best and second method for each metrics, respectively.[1]

| Model | Benchmark Results | | | | | Model Information | | | | |
|---|---|---|---|---|---|---|---|---|---|---|
| | Spearman | AUC | MCC | NDCG | Recall | Seq. | Struct. | MSA | # Params. | # Data |
| ProGen2 XL | 0.391 | 0.717 | 0.306 | 0.767 | 0.199 | | | | 6.4B | >1B |
| CARP | 0.368 | 0.701 | 0.285 | 0.748 | 0.208 | ✓ | ✗ | ✗ | 640M | 41M |
| ESM-2 | 0.414 | 0.729 | 0.327 | 0.747 | 0.217 | | | | 650M | 49M |
| DeepSequence | 0.419 | 0.729 | 0.328 | 0.776 | 0.226 | | | | 70M | N/A |
| MSA Transformer | 0.434 | 0.738 | 0.340 | 0.779 | 0.224 | | | | 100M | 26M |
| Tranception L | 0.434 | 0.739 | 0.341 | 0.779 | 0.220 | ✓ | ✗ | ✓ | 700M | 250M |
| EVE | 0.439 | 0.741 | 0.342 | 0.783 | 0.230 | | | | 240M | 250M |
| GEMME | 0.455 | 0.749 | 0.352 | 0.777 | 0.211 | | | | <1M | N/A |
| TranceptEVE L | 0.456 | 0.751 | 0.356 | 0.786 | 0.230 | | | | 940M | 250M |
| ProteinMPNN | 0.258 | 0.639 | 0.196 | 0.713 | 0.186 | | | | 2M | 25K |
| MIF | 0.383 | 0.706 | 0.294 | 0.743 | 0.216 | ✗ | ✓ | ✗ | 3M | 19K |
| ESM-IF | 0.422 | 0.730 | 0.331 | 0.748 | 0.223 | | | | 142M | 19K |
| MIF-ST | 0.383 | 0.717 | 0.310 | 0.765 | 0.226 | | | | 643M | 19K |
| ProtSSN | 0.442 | 0.743 | 0.351 | 0.764 | 0.226 | | | | 148M | 30K |
| SaProt | 0.457 | 0.751 | 0.359 | 0.768 | 0.233 | ✓ | ✓ | ✗ | 650M | 40M |
| S2F | 0.454 | 0.749 | 0.359 | 0.762 | 0.227 | | | | 6M | 30K |
| S3F | 0.470 | 0.757 | 0.371 | 0.770 | 0.234 | | | | 20M | 30K |
| ProtSST ($K$=2048) | 0.507 | 0.777 | 0.398 | 0.774 | 0.236 | | | | 110M | 18.8M |
| S2F-MSA | 0.487 | 0.767 | 0.381 | 0.790 | 0.240 | | | | 246M | 30K |
| S3F-MSA | 0.496 | 0.771 | 0.387 | 0.792 | 0.244 | ✓ | ✓ | ✓ | 260M | 30K |
| VenusREM | 0.518 | 0.783 | 0.404 | 0.770 | 0.244 | | | | 110M | 18.8M |
| AIDO Protein-RAG | 0.518 | 0.784 | 0.405 | 0.789 | 0.239 | | | | 16B | 1.2T |
| **EvoIF (Ours)** | 0.489 | 0.768 | 0.384 | 0.782 | **0.250** | ✓ | ✓ | ✗ | 76M | 30K |
| **EvoIF + GEMME (ensemble)** | **0.518** | **0.784** | **0.409** | **0.796** | 0.246 | | | ✓ | 76M | 30K |

## 3.3 ABLATION STUDY

**Profile type ablation.** We evaluate the contribution of different profile types through systematic ablation studies (Table 2). Starting from a baseline model without any profile (Spearman correlation: 0.454), we observe that adding the cross-family evolutionary inverse folding profile alone improves performance to 0.478, while adding the within-family structural evolutionary profile alone yields a smaller improvement to 0.462. The combination of both profiles achieves optimal performance (0.489), demonstrating their complementary nature and synergistic effect in capturing comprehensive biological information.

Table 2: Ablation of profile types on ProteinGym dataset

| Profile Type | | Metric | | | | |
|---|---|---|---|---|---|---|
| Inverse Folding | Structure | Spearman | AUC | MCC | NDCG | Recall |
| ✗ | ✗ | 0.454 | 0.749 | 0.359 | 0.762 | 0.227 |
| ✗ | ✓ | 0.462 | 0.753 | 0.365 | 0.770 | 0.234 |
| ✓ | ✗ | 0.478 | 0.761 | 0.376 | 0.779 | 0.248 |
| ✓ | ✓ | **0.489** | **0.768** | **0.384** | **0.782** | **0.250** |

**Data ablation.** We evaluate our model's performance with varying training set sizes through random deletion to assess data efficiency. As shown in Figure 2(f), reducing training data impacts performance, demonstrating that training data quantity remains crucial for protein fitness prediction.

However, our method achieves competitive performance with only 30K samples compared to state-of-the-art methods that require 1.2T training samples (AIDO Protein-RAG-16B) or 18.8M samples (VenusREM). This efficiency stems from our model's ability to effectively integrate evolutionary information from homologous constraints and structural constraints, enabling more efficient learning from limited data, with training time costs reduced by up to $10^9$-fold (Figure 3).

**Homology quantity ablation.** As shown in Figure 2(e), we evaluate the impact of homologous sequence quantity by progressively and randomly reducing the number of available sequences. The results indicate that model performance depends on the number of homologous sequences, although the effect is not pronounced. These findings demonstrate the importance of homologous sequence availability for protein fitness prediction. The results also demonstrate the capability of our method to maintain competitive performance even when homologous sequences are limited.

## 3.4 ANALYSIS

Our method achieves superior performance across all tested scenarios, confirming that the structure-evolution joint representations are highly conserved and universal, with strong inductive biases that effectively compensate for limited evolutionary information, enabling accurate prediction of novel protein families. For a detailed qualitative analysis on a representative system, please refer to the case study in Appendix B. Additional analyses are shown in Appendix E.

We observe consistent performance improvements as the model progressively incorporates multi-scale protein features. Figure 2(a-d) presents performance comparisons grouped by function type, MSA depth, taxon, and mutation depth:

**Function type:** Our model demonstrates particularly strong performance in capturing organismal fitness and protein stability. For organismal fitness prediction, our method's superior performance stems from its ability to capture evolutionary relationships between different organisms and distinguish functional constraints across species. For protein stability prediction, our model's effectiveness arises from the direct relationship between protein structure and stability. While baseline methods (S2F, S2F-MSA) also incorporate structural information, our fundamental advantage lies in more comprehensive and efficient evolutionary encoding and representation capabilities, whereas sequence-based pLMs such as ESM-2 show clear limitations in capturing structure-related fitness effects.

---

[1]Parameter counts refer to trainable parameters only. For methods using frozen pre-trained models (e.g., S2F, S3F, EvoIF use frozen ESM-2-650M and/or ProteinMPNN), only trainable components are counted.

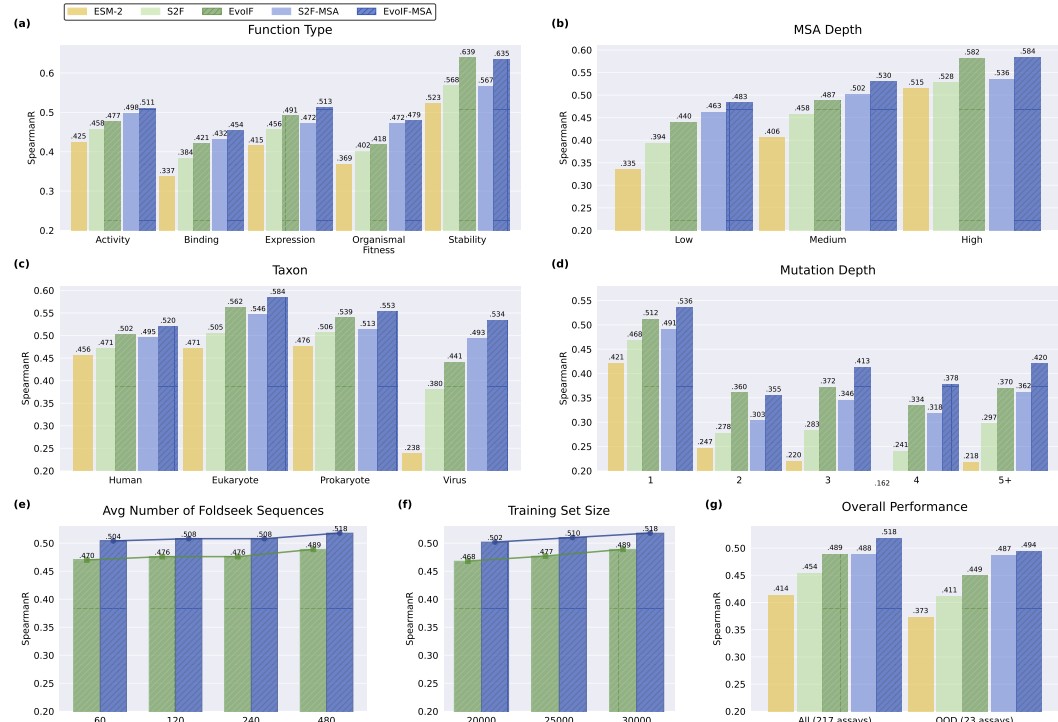

Figure 2: **Breakdown analysis** on ProteinGym, across **(a)** function type, **(b)** MSA depth, **(c)** taxon, and **(d)** mutation depth. **Ablation study** on **(e)** homology quantity and **(f)** training data size. **(g) Overall performance** on all assays and out-of-distribution assays.

**MSA depth:** Sequence-only methods suffer from reduced performance at low MSA depths due to weak evolutionary signals. By contrast, our method provides a more efficient encoding of evolutionary information and achieves superior performance as MSA depth increases, effectively capturing conservation, co-variation, and mutational tolerance, while also retaining informative patterns in deep MSAs.

**Taxon:** For underrepresented taxonomic group such like viruses, sequence-only models show reduced generalization capability due to taxonomic bias. This is because different viral families are often separated by larger evolutionary sequence distances. The sparsity of both known evolutionary sequences and experimental crystal structures for viruses contributes to this performance gap. However, our model still demonstrates performance improvements for viruses, indicating that our efficient evolutionary encoding and structural inductive biases can effectively compensate for insufficient data.

**Mutation depth:** As the number of mutated sites increases, the performance of all methods declines due to the limitations of the additive mutation effect assumption. In contrast, our method remains more stable and outperforms other approaches at 2, 3, 4, and even ≥5 mutations, indicating a superior ability to capture non-linear mutational interactions (epistasis).

**Generalizing to novel protein families.** While large-scale pLMs such as ESM-2 are pre-trained on massive sequence datasets like UniRef100, our methods (EvoIF and EvoIF-MSA) are trained on a much smaller dataset, using only 0.15% of the training data compared to large-scale models (Figure 3). A critical question arises: can the advantages of our methods generalize to protein families not seen during training? Figure 2(g) shows that in 23 out-of-distribution ProteinGym assays with low similarity to training data, all models exhibit performance degradation. However, our EvoIF and EvoIF-MSA methods consistently and significantly outperform the sequence-only baseline ESM-2. Moreover, our models also show a remarkable improvement over other baselines, demonstrating a superior ability to integrate both within-family evolutionary information from homolog profiles and cross-family inverse folding likelihood profiles for more accurate predictions. Detailed out-of-distribution evaluation results are provided in Appendix E.2.

## 4 DISCUSSION AND CONCLUSION

In this paper, we introduce EvoIF, a lightweight and data-efficient framework for protein fitness prediction that unifies two perspectives: an IRL-based interpretation of pLM zero-shot scoring, and a compact integration of within-family evolutionary information from homolog profiles with cross-family inverse folding likelihood profiles. Extensive evaluation on ProteinGym demonstrates that EvoIF and ts MSA-enabled variant EvoIF-MSA achieve state-of-the-art or competitive performance across 217 DMS assays while using only a fraction of the training data and parameters required by recent large-scale models. Ablations verify that the two profile sources are complementary, improving robustness across function types, MSA depths, taxa, and mutation depths.

This work highlights three takeaways. First, viewing MLM pretraining through the lens of inverse reinforcement learning clarifies why pLM log-odds correlate with fitness and motivates principled zero-shot scoring. Second, a compact evolutionary representation that combines sequence- and structure-retrieved homolog profiles with inverse folding profiles provides strong and uniformly available signals, mitigating the limitations of homolog searches in terms of limited scope and high computational cost. Third, a simple fusion via transition blocks suffices to yield calibrated probabilities for accurate log-odds estimation, obviating heavy model scaling.

Limitations include the fixed-backbone assumption and potential biases from structure availability. Future work will incorporate side-chain modeling, extend IRL formulation to handle epistasis, and explore joint training of sequence–structure backbones with profile encoders. Diffusion-based design priors and inference-time retrieval adaptation are promising directions for enhanced generalization.

## REPRODUCIBILITY STATEMENT

We have made extensive efforts to ensure the reproducibility of our work. Training details are provided in Appendix D.1, hyper-parameter settings in Appendix D.2, and homology retrieval details in Appendix D.3. For completeness, the main paper concisely details EvoIF's sequence–structure backbone, compact profile transition block, integration of (i) within-family homolog profiles and (ii) cross-family inverse folding profiles, and the training and inference procedures. Upon acceptance, we will release our models, together with training and inference code, to facilitate replication and further research.

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

# A   IMPACT OF MODEL AND DATA SCALE ON PROTEIN FITNESS PREDICTION PERFORMANCE

We summarize how accuracy (Spearman) varies with model parameter count and pre-training data scale. As shown in Figure 3, scaling parameters or data yields limited marginal gains for protein fitness prediction relative to computational cost, which aligns with our design that emphasizes compact evolutionary representations and efficient fusion in EvoIF-MSA.

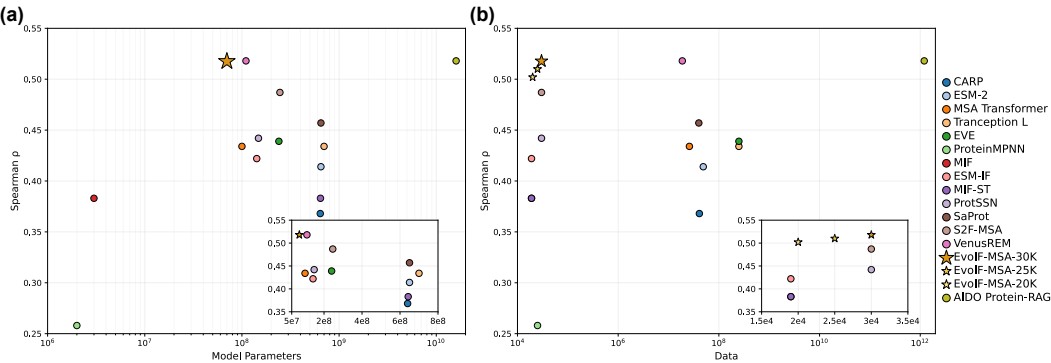

Figure 3: Accuracy (Spearman) versus **(a)** model parameters and **(b)** training data scale.

# B   CASE STUDY

Predicting the fitness of viral proteins is an important scientific problem. It enables the early identification of potential epidemiologically advantageous variants and accelerates the development of precise therapeutic strategies. In addition, accurate fitness prediction is highly valuable for engineering beneficial viruses such as bacteriophages. However, since different viruses are often separated by large evolutionary distances, the available within-family evolutionary information for viral proteins is usually limited. As a result, predicting the fitness of viral proteins has long been a challenge, and existing methods have struggled to achieve strong performance.

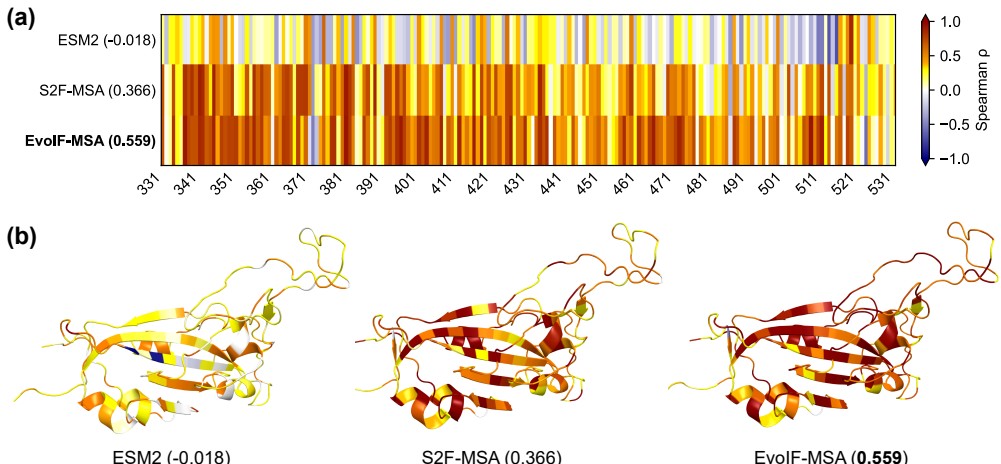

Figure 4: Visualization of fitness prediction results for the Spike glycoprotein. **(a)** Heatmap of per-site Spearman correlation coefficients of fitness prediction by ESM2-650M, S2F-MSA, and EvoIF-MSA. **(b)** Three-dimensional structure colored by per-site Spearman correlation coefficients of fitness prediction from ESM2-650M, S2F-MSA, and EvoIF-MSA. The structure was obtained from the ProteinGym database.

By explicitly modeling cross-family evolutionary information, our model achieves a significant improvement in viral fitness prediction (Figure 2). We select the Spike glycoprotein as a case study for analysis. This protein is essential for host cell recognition and membrane fusion and represents a central target for vaccine design and antibody neutralization. We compare our method with several baselines. The Spearman correlation coefficients of the sequence-based ESM2-650M model, the structure-based S2F-MSA model, and the evolution-based EvoIF-MSA model are -0.018, 0.366, and 0.559, respectively. These results demonstrate that EvoIF-MSA provides substantially more accurate fitness prediction. We further analyze the Spearman correlation coefficients of fitness prediction for different mutants at individual sites (Figure 4). EvoIF-MSA is able to better capture the mutational effects at sites that are structurally close but lack sufficient within-family evolutionary information. This highlights the advantage of EvoIF-MSA in providing a more comprehensive evolutionary profile for viral proteins.

## C  RELATED WORK

### C.1  PROTEIN FITNESS PREDICTION

Protein fitness prediction is a core task for understanding mutational effects and enabling rational protein design. Methodological progress largely tracks which biological signals are modeled and how they are combined.

Alignment-dependent approaches constitute the earliest paradigm. Models such as EVE [34], GEMME [16], and DeepSequence [7] extract position-specific statistics and co-evolutionary couplings from Multiple Sequence Alignments (MSAs). These methods work well when deep, high-quality MSAs exist but degrade for proteins with sparse homologs.

Large-scale protein language models (pLMs) introduced a family-agnostic alternative. Trained with masked language modeling (MLM) on massive sequence corpora, models such as ESM-2 [20], ProGen2 XL [27], and CARP-640M [49] achieve strong zero-shot estimation of mutational effects via log-odds scoring, without labeled fitness supervision. This capability provides a robust baseline across diverse families.

Structure-informed approaches leverage 3D constraints to improve robustness and biological plausibility. ProteinMPNN [2], MIF [48], and ESM-IF [14] demonstrate that incorporating geometric inductive biases benefits fitness prediction, especially for structure-sensitive properties. Hybrid sequence–structure models, including ProSST [17], ProtSSN [43], and S2F/S3F [50], further enhance accuracy in MSA-free settings. Complementarily, MSA-enhanced hybrids such as MSA Transformer [33], Tranception and TranceptEVE [28, 29] combine family-agnostic pLMs with family-specific alignment signals. Recent systems like VenusREM [42] and AIDO-Protein-RAG [41, 18] highlight the value of jointly exploiting structural and evolutionary information.

Collectively, these lines of work show that accurate fitness prediction benefits from integrating complementary signals: sequence statistics (pLMs), structural constraints (inverse folding and geometry-aware backbones), and within-family evolutionary couplings (MSAs or profiles). They also expose limitations—heavy reliance on data/model scale, sensitivity to MSA depth, and fragmented use of evolutionary information—motivating lightweight, unified approaches. EvoIF targets this gap by combining within-family homolog profiles with cross-family structural–evolutionary priors from inverse folding in a compact fusion framework.

### C.2  INVERSE REINFORCEMENT LEARNING

Inverse Reinforcement Learning (IRL) infers a reward function from expert demonstrations rather than optimizing actions for a given reward. In Maximum Entropy IRL, expert behavior is modeled by a Boltzmann distribution over trajectories proportional to cumulative reward [26, 52]. Viewing protein evolution as a sequential decision process, natural selection acts as the expert that preferentially retains high-fitness sequences. Under this lens, MLM on extant sequences resembles IRL: maximizing conditional log-likelihood aligns with maximizing an IRL objective on the expert's stationary distribution.

This correspondence implies that pLM log-probabilities provide an affine surrogate for reward; differences in log-probabilities (i.e., log-odds) approximate reward differences between mutant and wild-type, explaining the empirical success of zero-shot scoring used throughout the literature [24, 30]. Extending the analogy, incorporating homologous sequences—retrieved by sequence or structure similarity—can be interpreted as supplying additional expert demonstrations *in context*, sharpening reward inference for the local family neighborhood. This perspective provides a principled rationale for combining pLMs with evolutionary context and motivates EvoIF's use of both homolog profiles and inverse folding priors for calibrated log-odds estimation.

### C.3 Evolutionary Information Representation

Compact representations of evolutionary constraints have progressed from raw MSAs to profile-style and structure-aware surrogates. Classical alignment-based models use position-specific frequencies and co-evolutionary couplings derived from MSAs [4], but performance depends on family depth and retrieval quality. To improve scalability and uniformity, recent work in design and structure prediction emphasizes evolutionary profiles that summarize homolog statistics while remaining model-friendly [9, 32, 23]. Structure-centric retrieval (e.g., Foldseek) expands beyond sequence-detectable homology, stabilizing profiles in remote regimes [44, 42].

Inverse folding offers a complementary, cross-family source of evolutionary signal: structure-conditioned sequence recovery models assign high likelihoods to amino acids consistent with natural variation, thereby distilling structural–evolutionary couplings learned from broad protein space [37, 5]. These likelihoods function as informative, uniformly available priors, particularly valuable when MSAs are shallow, uneven, or expensive to retrieve. EvoIF integrates both sources—structure-retrieved homolog profiles and inverse folding likelihood profiles—through a lightweight transition block that fuses probabilities from sequence–structure backbones with compact evolutionary profiles. This design yields calibrated log-odds scoring while avoiding the computational cost and non-uniformity of deep homolog searches.

## D Implementation Details

### D.1 Training Details

During pre-training, we randomly select 15% of the residues in each protein sequence and apply the following token modification scheme: 80% of the selected residues are replaced with a `[MASK]` token, 10% are swapped with a random residue token, and the remaining 10% are left unchanged. The model is then tasked with predicting the original, unmodified residue.

The weights of the ESM-2-650M and ProteinMPNN models are frozen, with only the profile transition blocks for the external profiles and the GVP layers for the structure graphs remaining trainable. We train our model on four NVIDIA H800 GPUs for 80 epochs, which takes approximately 5 hours. Empirically, a mini-batch size of 32 per GPU (128 in total) yields better representation quality than 64 or 128 per GPU, so we keep this setting throughout our experiments.

### D.2 Hyper-parameters

We employ a hybrid optimizer that combines Muon [21] for matrix parameters and AdamW [22] for other parameters. Matrix parameters (defined as parameters with dimensionality $\geq$2D) are optimized using Muon with a learning rate of $1 \times 10^{-3}$, momentum of 0.95, 5 Newton-Schulz steps, and weight decay of 0.1. The remaining parameters use AdamW with $\beta_1 = 0.9$, $\beta_2 = 0.95$, $\epsilon = 1 \times 10^{-8}$, and weight decay of 0.1.Parameters are automatically routed based on dimensionality, with Muon learning rates scaled by matrix dimensions to ensure stable convergence.

### D.3 Homology Retrieval

We performed homology searches using Foldseek [44] against the AlphaFold Proteome database, a curated subset derived from the full AlphaFold Protein Structure Database [45] that contains high-confidence predicted structures for complete proteomes of key model organisms. To enable sensitive remote homology detection, we employed Foldseek with high-sensitivity settings (sensitivity: 9.5) in

structural alignment mode (3Di+AA). We applied a maximum sequence identity cutoff of 90% to reduce redundancy, resulting in an average of approximately 500 homologous sequences per query. The resulting alignments in A3M format were subsequently processed by realigning all sequences to the query length via truncation or padding while preserving gap characters ("-"). We then construct the position-specific profile $P$ directly from the aligned homologs following Equation 8 and use it as the evolutionary prior in our fusion module.

## D.4 EVALUATION METRICS

To comprehensively evaluate the performance of protein fitness prediction, we employ a set of five metrics: (1) Spearman's rank correlation coefficient (**Spearman**), which quantifies the monotonic relationship between model-predicted fitness scores and experimentally measured values, effectively capturing ordinal agreement without assuming linearity. (2) The area under the receiver operating characteristic curve (**AUC**) assesses binary classification performance across varying discrimination thresholds. (3) Matthews correlation coefficient (**MCC**) evaluates classification quality in the presence of class imbalance, offering a balanced perspective on prediction accuracy. (4) Normalized discounted cumulative gain (**NDCG**) measures the model's capability to correctly rank highly functional variants. (5) Top-10% recall (**recall**) calculates the proportion of truly functional mutants identified within the top decile of model predictions. All metrics are computed using standardized scripts from the ProteinGym repository to ensure reproducibility and consistency with established benchmarks.

## D.5 MODEL ARCHITECTURE

Figure 5 illustrates the Geometric Sequence-Structure Encoder component of EvoIF. Specifically, ESM features are used to initialize the node features within the Geometric Sequence-Structure Encoder (GNN). Beyond the GVP-GNN architecture shown in the figure, EvoIF incorporates two types of evolutionary profiles: (1) Structure Profile ($P^{\text{struct}}$) derived from within-family structural homologs retrieved via Foldseek, and (2) Inverse Folding Profile ($P^{\text{IF}}$) obtained from ProteinMPNN, which provides cross-family structural–evolutionary constraints. Both profiles are processed through separate transition blocks (transformer layers) and then combined with the GNN output via addition at the logits level, as detailed in Equation 9.

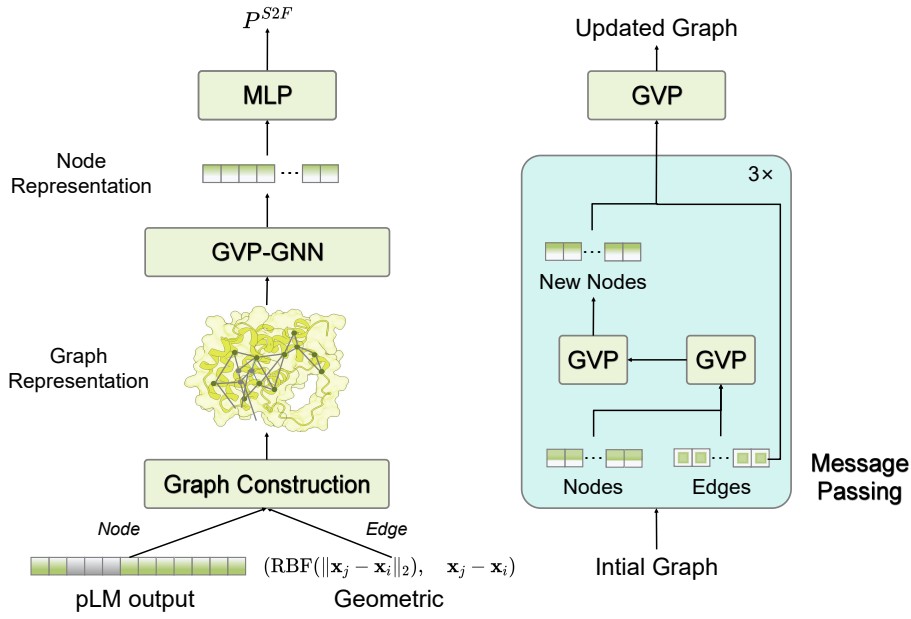

Figure 5: Geometric Sequence-Structure Encoder architecture of EvoIF.

# E ADDITIONAL ANALYSES

We present additional analysis of EvoIF's performance across different protein function types and experimental conditions on the ProteinGym benchmark.

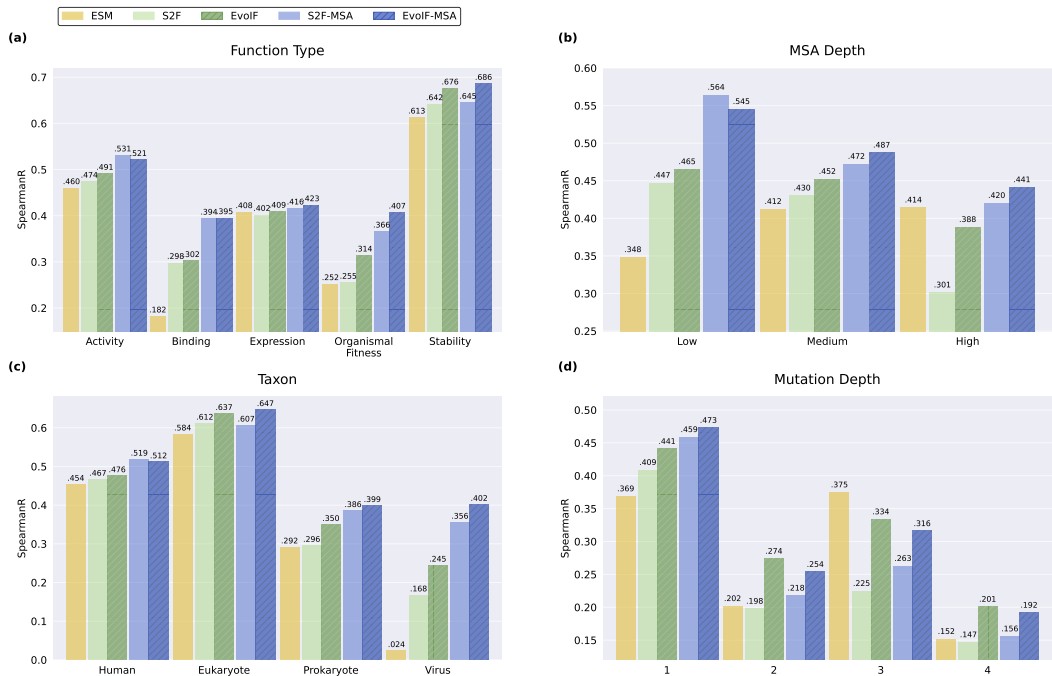

Figure 6: Out-of-distribution evaluation on 23 ProteinGym assays with low similarity to training data, across **(a)** Function Type, **(b)** MSA Depth, **(c)** Taxon, and **(d)** Mutation Depth. EvoIF and EvoIF-MSA maintain superior Spearman correlation compared to sequence-only and prior sequence–structure baselines.

## E.1 DETAILED PERFORMANCE ACROSS FUNCTION TYPES

We report per-assay Spearman correlations for activity assays (Figure 7), organismal fitness assays (Figure 8), stability assays (Figure 9), expression assays (Figure 10), and binding assays (Figure 11).

## E.2 OUT-OF-DISTRIBUTION EVALUATION

Figure 6 shows the out-of-distribution evaluation results of EvoIF and EvoIF-MSA on 23 ProteinGym assays with low similarity to the training data. The results show that our approach consistently achieves superior performance under Out-of-distribution conditions, which highlights the strong generalization ability of EvoIF and EvoIF-MSA. The advantage is particularly evident for viral proteins, as they exhibit greater evolutionary heterogeneity. Viral families with similar functions often have low sequence similarity but share similar structural features. As a result, our explicit modeling of cross-family structural evolutionary information significantly improves the model's ability to capture comprehensive evolutionary signals. In addition, our method more effectively captures fitness effects across different mutation depths, which underscores its ability to model epistatic interactions associated with multiple mutations.

## E.3 ALTERNATIVE INVERSE FOLDING MODELS: ESM-IF AND CALIBY

To demonstrate that the effectiveness of inverse folding logits is not specific to ProteinMPNN, we evaluated our method using alternative inverse folding models: ESM-IF and Caliby. As shown

in Table 3, all three inverse folding models (ProteinMPNN, ESM-IF, and Caliby) show consistent improvements when incorporating MSA ensemble, confirming that the benefits of using inverse folding logits stem from capturing evolutionary priors rather than being model-specific.

Table 3: Performance comparison across different inverse folding models (ProteinMPNN, ESM-IF, and Caliby) with and without MSA ensemble.

| Inverse Folding Model | MSA | Spearman | AUC | MCC | NDCG | Top-recall |
|---|---|---|---|---|---|---|
| ProteinMPNN | ✗ | 0.489 | 0.768 | 0.384 | 0.782 | 0.250 |
| | ✓ | 0.518 | 0.784 | 0.409 | 0.796 | 0.246 |
| ESM-IF | ✗ | 0.481 | 0.764 | 0.381 | 0.778 | 0.243 |
| | ✓ | 0.513 | 0.781 | 0.408 | 0.792 | 0.244 |
| Caliby | ✗ | 0.459 | 0.752 | 0.359 | 0.769 | 0.230 |
| | ✓ | 0.496 | 0.773 | 0.392 | 0.787 | 0.231 |

### E.4 IMPACT OF HOMOLOGOUS SEQUENCE SIMILARITY THRESHOLD

To ensure that FoldSeek-retrieved homologs are within the same protein family and share similar evolutionary constraints, we conducted ablation studies varying the minimum sequence similarity threshold (0%, 20%, 30%, 40%, 50%). As shown in Table 4, model performance remains stable across different similarity thresholds, indicating that our method effectively utilizes structurally similar proteins while maintaining evolutionary relevance. The results demonstrate that FoldSeek's structural similarity search successfully identifies evolutionarily related proteins even at low sequence similarity levels.

Table 4: Impact of homologous sequence similarity threshold on model performance. Results are reported for configurations with and without MSA ensemble.

| MSA | Threshold | Spearman | AUC | MCC | NDCG | Top-recall |
|---|---|---|---|---|---|---|
| | 0.0 | 0.518 | 0.784 | 0.409 | 0.796 | 0.246 |
| | 0.2 | 0.513 | 0.781 | 0.404 | 0.796 | 0.247 |
| ✓ | 0.3 | 0.515 | 0.782 | 0.406 | 0.793 | 0.244 |
| | 0.4 | 0.512 | 0.780 | 0.403 | 0.793 | 0.245 |
| | 0.5 | 0.510 | 0.780 | 0.401 | 0.792 | 0.242 |
| | 0.0 | 0.489 | 0.768 | 0.384 | 0.782 | 0.250 |
| | 0.2 | 0.482 | 0.764 | 0.379 | 0.785 | 0.246 |
| ✗ | 0.3 | 0.484 | 0.764 | 0.381 | 0.780 | 0.242 |
| | 0.4 | 0.481 | 0.763 | 0.378 | 0.777 | 0.241 |
| | 0.5 | 0.480 | 0.763 | 0.376 | 0.778 | 0.239 |

### E.5 INFERENCE TIME ANALYSIS

We provide a comprehensive analysis of inference time for different components and methods. Table 5 reports the time required for FoldSeek homology search in the AlphaFold Database and ProteinMPNN inverse folding computation. Table 6 shows the MSA computation time for VenusREM on different proteins. Table 7 compares the total inference time across different methods on the

ProteinGym benchmark, demonstrating that our method achieves competitive performance with reasonable computational overhead.

Table 5: Inference time for FoldSeek homology search in the AlphaFold Database and ProteinMPNN inverse folding computation.

| Component | Dataset | # Proteins | Hardware | Inference Time |
|---|---|---|---|---|
| FoldSeek | CATH | 30,948 | 64 CPU cores | 33 min 45 sec |
|  | ProteinGym | 217 | 64 CPU cores | 71 sec |
| ProteinMPNN | CATH | 30,948 | 1 H800 GPU, 64 CPU cores | 7 min 43 sec |
|  | ProteinGym | 217 | 1 H800 GPU, 64 CPU cores | 10 sec |

Table 6: MSA computation time for different proteins (96 CPUs). We selected several representative cases for analysis.

| Protein | Sequence Length | Time |
|---|---|---|
| YNZC_BACSU | 39 | 5h 18m |
| VKOR1_HUMAN | 163 | 5h 1m |
| Q6wV13_9MAXI | 222 | 4h 47m |
| C6KNH7_9INFA | 566 | 5h 11m |

Table 7: Total inference time comparison across different methods on the ProteinGym benchmark (excluding MSA recomputation time).

| Method | Dataset | Inference Time |
|---|---|---|
| VenusREM | ProteinGym | 3h 6m 36s |
| S2F | ProteinGym | 1h 4m 58s |
| S3F | ProteinGym | 6h 53m 48s |
| EvoIF | ProteinGym | 1h 12m 6s |

## E.6 ARCHITECTURE ABLATION: GVP VS GEARNET

To validate our choice of GVP as the structure encoder, we conducted an ablation study comparing GVP with GearNet, another graph neural network architecture commonly used for protein structure modeling. As shown in Table 8, while both architectures benefit from incorporating MSA ensemble, GVP consistently outperforms GearNet across all metrics. This finding aligns with the evaluation reported in Zhang et al. [50], confirming that GVP is more effective for fitness prediction tasks.

Table 8: Performance comparison between GVP and GearNet architectures with and without MSA ensemble.

| Model | MSA | Spearman | AUC | MCC | NDCG | Top-recall |
|---|---|---|---|---|---|---|
| GVP | ✗ | 0.489 | 0.768 | 0.384 | 0.782 | 0.250 |
| | ✓ | 0.518 | 0.784 | 0.409 | 0.796 | 0.246 |
| GearNet | ✗ | 0.473 | 0.758 | 0.371 | 0.771 | 0.237 |
| | ✓ | 0.508 | 0.777 | 0.397 | 0.792 | 0.242 |

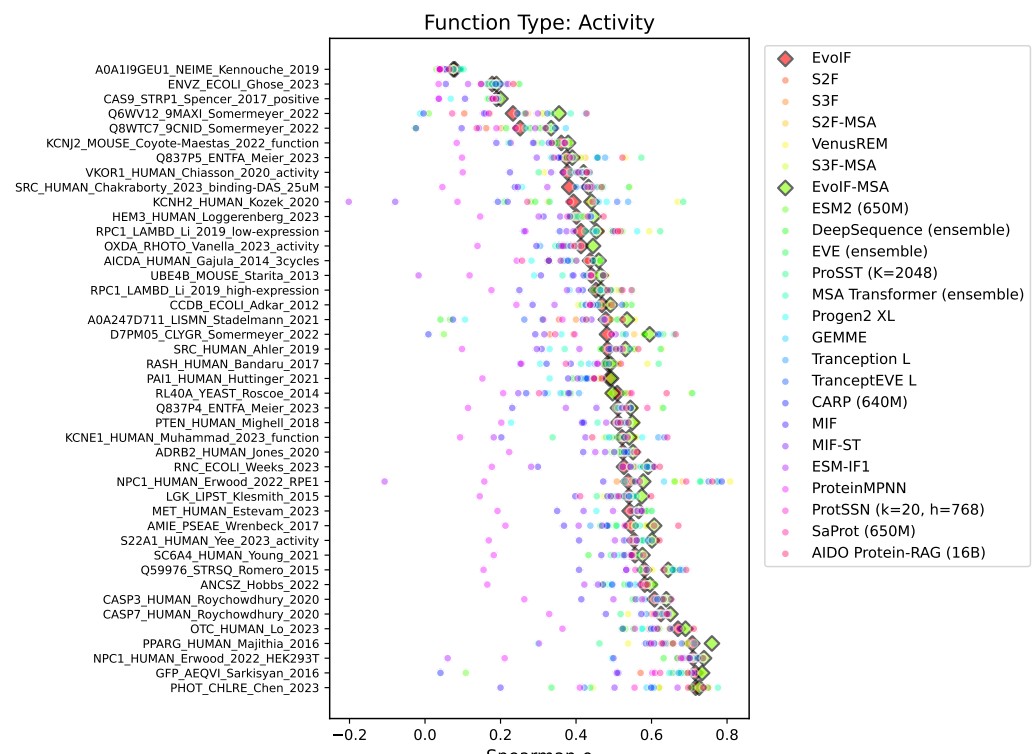

Figure 7: Per-assay Spearman correlation for activity assays on ProteinGym.

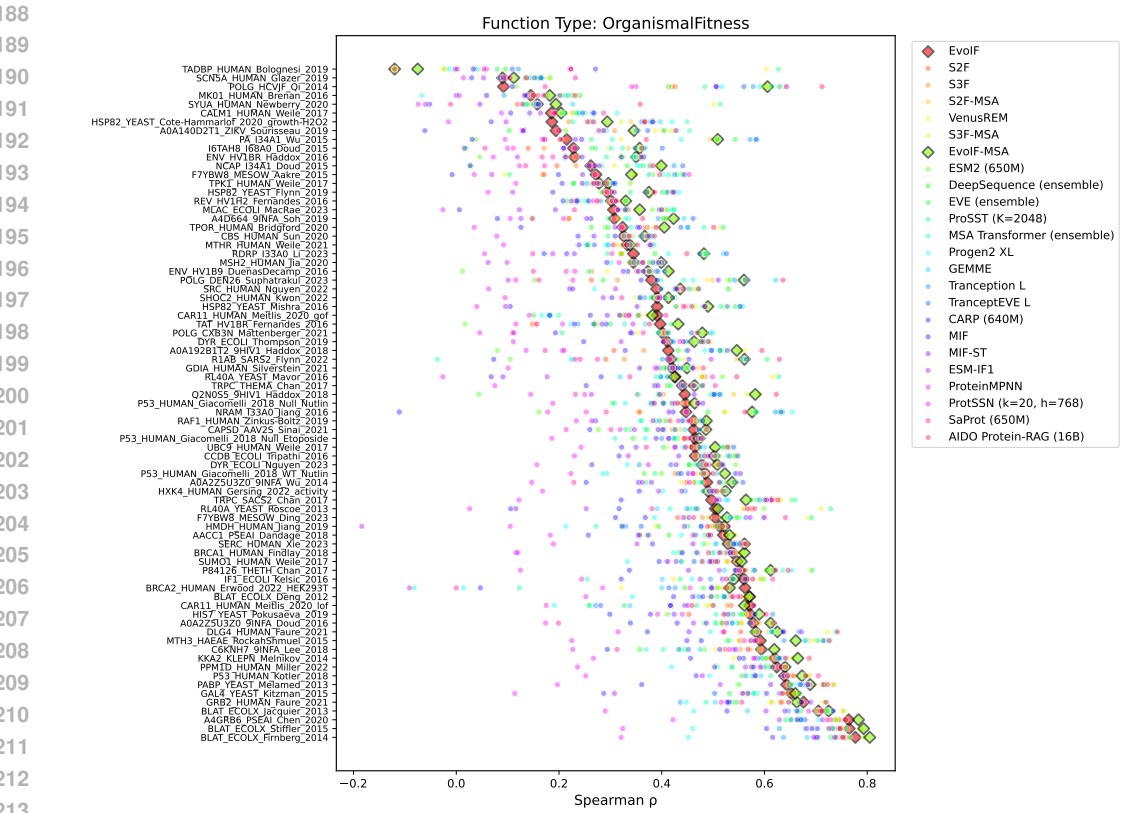

Figure 8: Per-assay Spearman correlation for organismal fitness assays on ProteinGym.

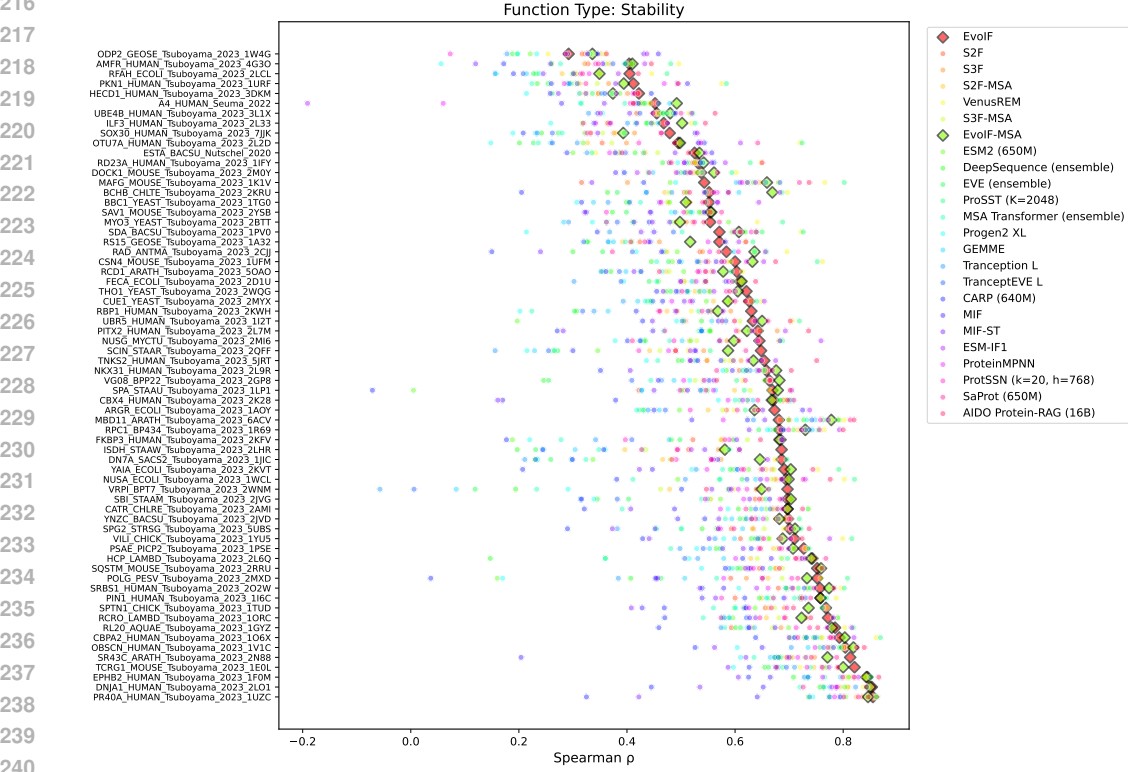

Figure 9: Per-assay Spearman correlation for stability assays on ProteinGym.

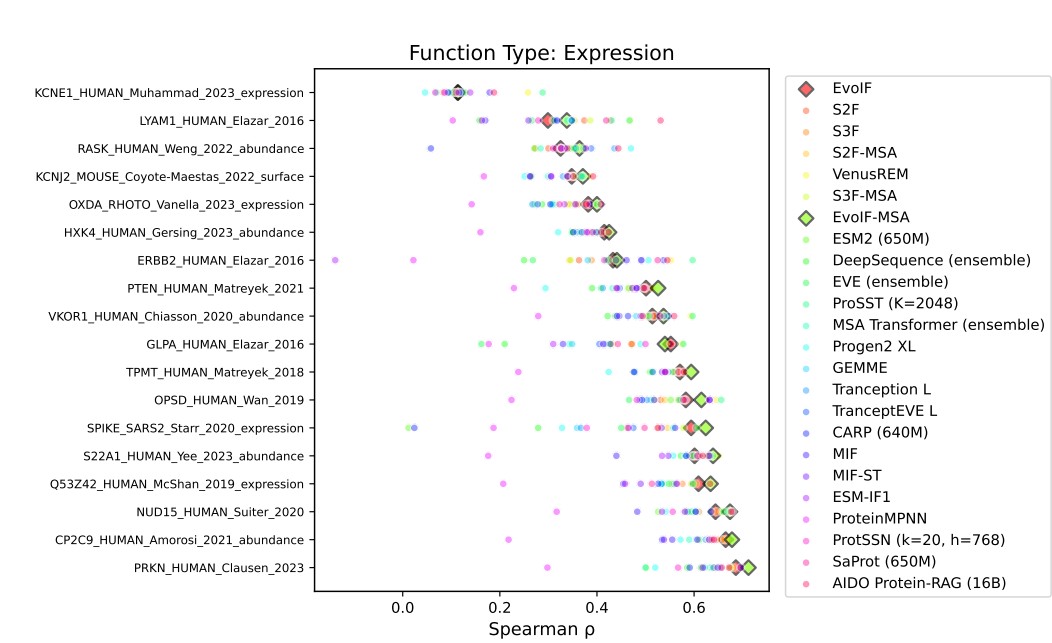

Figure 10: Per-assay Spearman correlation for expression assays on ProteinGym.

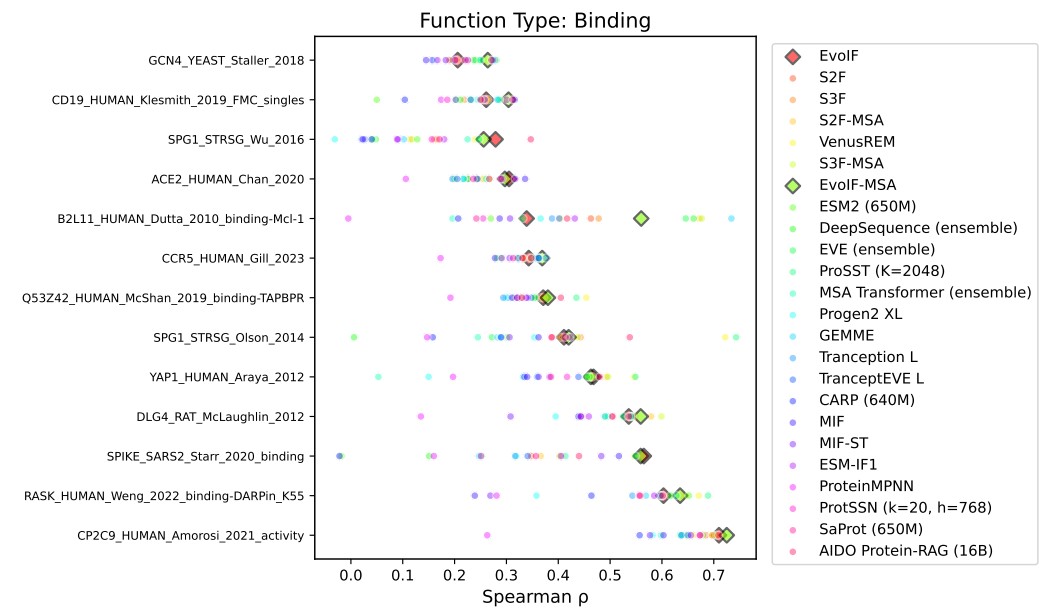

Figure 11: Per-assay Spearman correlation for binding assays on ProteinGym.

## ETHICS STATEMENT

This work adheres to the ICLR Code of Ethics. In this study, no human subjects or animal experimentation was involved. All datasets used, including ProteinGym and CATH, were sourced in compliance with relevant usage guidelines, ensuring no violation of privacy. We have taken care to avoid any biases or discriminatory outcomes in our research process. No personally identifiable information was used, and no experiments were conducted that could raise privacy or security concerns. We are committed to maintaining transparency and integrity throughout the research process.

## LLM USAGE STATEMENT

LLMs assist with translation and stylistic editing to enhance clarity and grammatical precision. All LLM-generated outputs undergo rigorous verification against authoritative sources, particularly for technical terminology and nuanced translations.

