# OpenReview forum: "Evolutionary Profiles for Protein Fitness Prediction"
_ICLR.cc/2026/Conference — Submitted to ICLR 2026_

### Official Review · Reviewer_uK6d · 2025-10-29

**Soundness:** 3
**Presentation:** 3
**Contribution:** 2
**Rating:** 4
**Confidence:** 4

**Summary:**

This paper introduces EvoIF, which takes within- and cross-family evolutionary profiles and structural information as inputs to model protein fitness. It reaches state-of-the-art with 76M params and 30k training set. The main contribution of the paper includes:
1. Unified IRL interpretation of pLMs: The paper reinterprets MLM in pLMs as a form of inverse reinforcement learning, where log-likelihoods implicitly approximate evolutionary rewards.
2. Structure-sequence-evolutionary profiles. EvoIF combines a GVP encoder to process protein structure and and two complementary evolutionary profiles (within- and cross-family) through a lightweight transition block.

**Strengths:**

There are several strengths of this paper:
1. The paper provides a unified theoretical framing by interpreting MLM as IRL, where plms implicitly estimate evolutionary “rewards.” While I'm not expert in RL and IRL so I will look for other reviewers' opinions about this.
2. The authors construct within-family and cross-family profiles and capture distinct and complementary signals. The authors also use ablation study to prove the effectiveness of different profiles.

**Weaknesses:**

However, there are several issues with these paers:
1. As I said in strengths, I will look for other reviewers' opinions about regarding MLM and IRL.
2. The comparison of parameter and training dataset size isn't fair enough. Since EvoIF requires homologs retrieval, run plm and inverse folding model. The authors should also compare the run time instead of comparing the parameter and dataset only.
3. Limited novelty in model architecture. The structure embedding, profiles fusion module are incremental combinations of existing tools. I know there is not a lot of things could be done with the model architecture during the rebuttal stage, but the authors can think about this problem.

There are some over-claims in this paper:
1. The authors claim that the inverse folding captures the cross-family evolutionary profile. While none of previous inverse folding paper have explained this as "cross-family evolutionary" information. These information is more likely to be biophysical prior. Claiming it as cross-family evolutionary information may mislead the audience.
2. The claim "Existing models have not fully considered the comprehensive modeling of protein evolutionary information" is not true. At least LM-Design[1] already utilized evolutionary information from protein language model to assist protein structure to sequence design.

[1] Zheng, Z., Deng, Y., Xue, D., Zhou, Y., Ye, F., & Gu, Q. (2023, July). Structure-informed language models are protein designers. In International conference on machine learning (pp. 42317-42338). PMLR.

**Questions:**

1. A comparison of run time will be beneficial in evaluating different methods.
2. Reduce the over-claims or make stronger proof of the claim.

---

> ### Author Response · Authors · 2025-11-24
>
> Dear Reviewer:
>
> Thank you for your thorough review and constructive comments! We sincerely hope our responses fully address your questions, and we would greatly appreciate it if you could consider raising your rating.
>
> ---
>
> > **W1.** As I said in strengths, I will look for other reviewers' opinions about regarding MLM and IRL.
>
> **A:**
> We appreciate the reviewer's concern.
>
> We want to clarify that our IRL section does not propose a new optimization algorithm, but rather provides a unifying theoretical framework to rigorously justify why MLM-based log-odds scoring works for fitness prediction. By framing evolution as an MDP and MLM as Pseudo-Likelihood maximization, we bridge the gap between "black-box" pre-training and evolutionary applicability.
>
> We are encouraged that other reviewers found this perspective valuable:
>
> > Reviewer e155 states: `The IRL interpretation is elegant and clarifies why likelihood-based models correlate with evolutionary fitness.`
>
> > Reviewer ewoj notes that our approach `provides a conceptual justification for using pLM log-odds as fitness estimators.`
>
> > Reviewer cA4c highlights that `The parallel between protein evolution and RL is interesting.`
>
>
> ---
>
> > **W2.** The comparison of parameter and training dataset size isn't fair enough. Since EvoIF requires homologs retrieval, run plm and inverse folding model. The authors should also compare the run time instead of comparing the parameter and dataset only.
>
> > **Q1.** A comparison of run time will be beneficial in evaluating different methods.
>
> **A:**
> We appreciate this important point. We provide inference time comparison in Tables R1, R2, and R3. EvoIF inference time is competitive with other structure-aware methods (comparable to S2F, and faster than S3F and VenusREM) and significantly faster than MSA-based approaches. Component-level analysis shows that FoldSeek retrieval and ProteinMPNN computation add minimal overhead compared to MSA-based methods, demonstrating the efficiency advantage of our structure-based retrieval. The performance gains justify this modest computational cost.
>
> **Table R1:** Inference time for FoldSeek homology search in the AlphaFold Database and ProteinMPNN inverse folding computation.
>
> | Component | Dataset | # Proteins | Hardware | Inference Time |
> |-----------|---------|-----------|----------|----------------|
> | FoldSeek | CATH | 30,948 | 64 CPU cores | 33 min 45 sec |
> | FoldSeek | ProteinGym | 217 | 64 CPU cores | 71 sec |
> | ProteinMPNN | CATH | 30,948 | 1 H800 GPU, 64 CPU cores | 7 min 43 sec |
> | ProteinMPNN | ProteinGym | 217 | 1 H800 GPU, 64 CPU cores | 10 sec |
>
> **Table R2:** Total inference time comparison across different methods on the ProteinGym benchmark (excluding MSA recomputation time).
>
> | Method | Dataset | Inference Time |
> |--------|---------|----------------|
> | VenusREM | ProteinGym | 3h 6m 36s |
> | S2F | ProteinGym | 1h 4m 58s |
> | S3F | ProteinGym | 6h 53m 48s |
> | EvoIF | ProteinGym | 1h 12m 6s |
>
> **Table R3:** MSA computation time for different proteins (96 CPUs). We selected several representative cases for analysis.
>
> | Protein | Sequence Length | Time |
> |---------|----------------|------|
> | YNZC_BACSU | 39 | 5h 18m |
> | VKOR1_HUMAN | 163 | 5h 1m |
> | Q6wV13_9MAXI | 222 | 4h 47m |
> | C6KNH7_9INFA | 566 | 5h 11m |
>
> ---
>
> > **W3.** Limited novelty in model architecture. The structure embedding, profiles fusion module are incremental combinations of existing tools. I know there is not a lot of things could be done with the model architecture during the rebuttal stage, but the authors can think about this problem.
>
> **A:**
>
> We evaluated alternative GNN architectures in the supplementary material (Table 8), comparing GVP with GearNet, another commonly used graph neural network for protein structure modeling. GVP consistently outperforms GearNet across all evaluation metrics (0.489 vs 0.473 Spearman correlation without MSA, 0.518 vs 0.508 with MSA), validating our architectural choice. This finding aligns with previous evaluations in S2F and demonstrates that architectural complexity does not guarantee performance gains in fitness prediction. The key lies in principled design and effective integration of complementary components.
>
> ---

---

> ### Author Response · Authors · 2025-11-24
>
> > **W4.** The authors claim that the inverse folding captures the cross-family evolutionary profile. While none of previous inverse folding paper have explained this as "cross-family evolutionary" information. These information is more likely to be biophysical prior. Claiming it as cross-family evolutionary information may mislead the audience.
>
> **A:**
> Thank you for your insightful comment. We understand the concern about the terminology and appreciate the opportunity to clarify our intent. Our use of the term "cross-family evolutionary information" is not meant to make an over-claim, but rather to describe a phenomenon that is well recognized in protein biology—namely, that structural conservation can encode evolutionary constraints even when sequence similarity falls below conventional family-level thresholds.
>
> Although earlier inverse folding papers may not explicitly employ this term, closely related concepts have been discussed in prior biological work. For example, the recent Cell study [1] notes that inverse folding models capture information that extends beyond sequence-defined family boundaries, which aligns with the spirit of what we describe as "cross-family" information.
>
> Our motivation for using this term is to distinguish between two forms of evolutionary signal:
>
> * Within-family evolutionary information derived from high sequence similarity
>
> * Cross-family evolutionary information originating from conserved structural or functional constraints that persist despite substantial sequence divergence
>
> A representative biological example is the Single-Strand Annealing Protein (SSAP) superfamily [2,3], which includes three families—(1) Rad52-type, (2) RecT/Redβ-type, and (3) Erf-type. These proteins share the same SSAP function and highly similar core structures, yet their inter-family sequence identities fall below 30%. This illustrates how structural conservation can bridge relationships across sequence-divergent groups, consistent with our usage of the term.
>
> Furthermore, our experimental findings—particularly the viral protein case study in the Appendix—show that incorporating this structurally derived cross-family signal leads to measurable performance gains. These improvements appear not only over pure sequence models (e.g., ESM-2) but also beyond structure-aware models such as S2F, reinforcing our motivation for explicitly modeling this source of information.
>
> [1] Advancing protein evolution with inverse folding models integrating structural and evolutionary constraints. Cell 2025.
>
> [2] Mechanism of single-stranded DNA annealing by RAD52–RPA complex. Nature 2024.
>
> [3] The Rad52 SSAP superfamily and new insight into homologous recombination. Communications Biology 2023.
>
> ---
>
> > **W5.** The claim "Existing models have not fully considered the comprehensive modeling of protein evolutionary information" is not true. At least LM-Design[1] already utilized evolutionary information from protein language model to assist protein structure to sequence design.
> > **Q2.** Reduce the over-claims or make stronger proof of the claim.
>
> **A:**
> We clarify that our claim refers to the comprehensive modeling of evolutionary information through the fusion of multiple complementary sources. While LM-Design has utilized evolutionary information, its modeling approach differs: LM-Design uses IF information to assist structure-to-sequence design, whereas we fuse inverse folding profiles with structure-retrieved profiles in a unified framework for fitness prediction. To our knowledge, this comprehensive fusion approach has not been previously applied to the fitness prediction task. Our ablation studies demonstrate that both profile types contribute complementary information.

---

### Official Review · Reviewer_cA4c · 2025-10-31

**Soundness:** 3
**Presentation:** 3
**Contribution:** 3
**Rating:** 8
**Confidence:** 4

**Summary:**

The authors propose EvoIF, a framework for protein fitness prediction that combines three pathways that are combined using a fusion module to calculate mutation preferences. The authors also theoretically parallelize log-odd ratios to inverse RL objectives to justify their use and correlation results as a zero-shot predictor for fitness. Results for a variant of EvoIF using MSA achieve state-of-the-art performance for fitness prediction in ProteinGym.

**Strengths:**

1. The parallel between protein evolution and RL is interesting.
2. The proposed fusion module and encoder are simple but well-thought-out.
3. The proposed model achieves state-of-the-art performance in ProteinGym.

**Weaknesses:**

1. The method seems dependent on structure availability and homolog retrieval.
2. The technical novelty of the proposed method is limited.
3. Code is not available.

**Questions:**

The paper is well-written, well-thought-out, and the results are convincing. My initial recommendation is acceptance. My detailed comments are as follows.

Comments:

1. (lines 117-118) The affirmation of “using only 0.15% of the training data and fewer model parameters” is strange, given that ESM-2 with 650M parameters is an important part of EvoIF architecture to calculate node embeddings. This is also related to the number of parameters column in Table 1 that might need changes.
2. From my understanding, the proposed method is used as a zero-shot predictor using Equation 3. Does using Equation 3 is enough and generalize to proteins with different functions? Does the fitness need to be normalized in a specific manner? It might be important to discuss this fact, as I guess that this normalization is performed as a preprocessing step in ProteinGym.
3. What is the difference between your encoder and the encoder architecture proposed in Zhang et al?
4. In line 172 the authors mention that “fitness landscapes are assumed time-invariant”, which is a valid assumption. How do the authors think Equation 6 would change, assuming dynamic environments or time-dependence for protein evolution?
5. How long does it take to perform the inference for one protein with EvoIF-MSA? The bottleneck in this case seems to be the homolog retrieval and MSA calculation.

---

> ### Author Response · Authors · 2025-11-24
>
> Dear Reviewer:
>
> Thank you for your thorough review and insightful comments! In the following, we will address each issue in detail and provide the necessary experimental results.
>
> ---
>
> > **W1.** The method seems dependent on structure availability and homolog retrieval.
>
> **A:**
> We appreciate this concern. We note that dependence on structure availability and homolog retrieval is common across structure-aware fitness prediction methods.
> **Structure availability:** ProteinGym primarily uses AlphaFold2-predicted structures, demonstrating that structure availability is not fundamentally limited by experimental determination. Predicted structures have been widely validated in ProteinGym and show consistent gains over sequence-only methods.
> **Homolog retrieval:** Homology search is a natural and computationally efficient approach for incorporating evolutionary information.  We use FoldSeek for structural homology retrieval, which is computationally lightweight. Our ablation study shows the model maintains competitive performance even with limited homologous sequences.
>
> ---
>
> > **W2.** The technical novelty of the proposed method is limited.
>
> **A:**
> We respectfully disagree that the novelty is limited to component combination. While we utilize established architectures (GVP, Transformer), the core novelty lies in the theoretical unification and its architectural realization.
>
> Theoretical Perspective: We reframe protein fitness prediction not just as pattern matching, but as Inverse Reinforcement Learning (IRL). We provide a rigorous link between standard MLM training and evolutionary reward maximization via the Pseudo-Likelihood interpretation. This offers a missing theoretical justification for why log-odds scoring works as a zero-shot fitness proxy.
>
> Architectural Realization: This theoretical lens directly guided our design. By viewing natural selection as an "expert policy," we treat retrieved homologs as "contextual expert demonstrations" rather than simple input features. This motivated our specific Fusion Module design, which integrates "local expert demonstrations" (Within-Family Profiles) with "global structural priors" (Inverse Folding Profiles) to constrain the policy search space.
>
>
> ---
>
> > **W3.** Code is not available.
>
> **A:**
> Thanks for your suggestion. We are currently organizing and refactoring the code, and we plan to release our code upon publication.
>
> ---
>
>
>
> > **Q1.** (lines 117-118) The affirmation of "using only 0.15% of the training data and fewer model parameters" is strange, given that ESM-2 with 650M parameters is an important part of EvoIF architecture to calculate node embeddings. This is also related to the number of parameters column in Table 1 that might need changes.
>
> **A:**
> We appreciate this important suggestion. We acknowledge that our claim about "fewer model parameters" requires clarification since ESM-2-650M is an integral component of our architecture.
>
>
> In Table 1, the parameter count for EvoIF (76M) refers to **trainable parameters only** (profile transition blocks and GVP layers). The ESM-2-650M and ProteinMPNN weights are frozen during training (Appendix D.1), so they are not included. This counting convention aligns with standard practice in the field: methods like S2F/S3F (Zhang et al., 2024) similarly use frozen PLMs as feature extractors while reporting only trainable parameters.
> We have added a footnote to Table 1 to explicitly state this convention.
>
>
> ---

---

> ### Author Response · Authors · 2025-11-24
>
> > **Q2.1.** From my understanding, the proposed method is used as a zero-shot predictor using Equation 3. Does using Equation 3 is enough and generalize to proteins with different functions? Does the fitness need to be normalized in a specific manner? It might be important to discuss this fact, as I guess that this normalization is performed as a preprocessing step in ProteinGym.
>
> **A:** Thanks and addressed. Since our method operates in a zero-shot setting (no fitness labels used during training), all performance on ProteinGym's 217 DMS assays reflects true generalization. These assays span diverse protein functions (activity, stability, binding, organismal fitness, and expression), and as shown in Figure 2, our method achieves consistent improvements across all function types without function-specific modifications, demonstrating effective generalization without function-specific modifications.
>
> > **Q2.2.** Does the fitness need to be normalized in a specific manner? It might be important to discuss this fact, as I guess that this normalization is performed as a preprocessing step in ProteinGym.
>
> **A:**
> Indeed, normalization is performed as a preprocessing step in ProteinGym. Specifically, it centers and normalizes to the interval $(-1, 1)$. Additionally, fitness is inherently a normalized metric. Since our evaluation uses Spearman correlation, which measures ranking accuracy, normalization does not affect our results, as rank-order relationships are invariant under monotonic transformations.
>
> ---
>
> > **Q3.** What is the difference between your encoder and the encoder architecture proposed in Zhang et al?
>
> **A:**
> Our encoder architecture follows Zhang et al. (S2F), which uses GVP for structure encoding. As reported in Zhang et al., S2F evaluated different GNN architectures and found GVP to be most effective for fitness prediction. We confirmed this finding through preliminary ablation and adopt this established architecture.
>
>
> ---
>
> > **Q4.** In line 172 the authors mention that "fitness landscapes are assumed time-invariant", which is a valid assumption. How do the authors think Equation 6 would change, assuming dynamic environments or time-dependence for protein evolution?
>
> **A:**
> We appreciate this question regarding the extension to time-dependent fitness landscapes. Under dynamic environments, the reward function becomes time-dependent: $R_\theta: \mathcal{S} \times \mathcal{T} \to \mathbb{R}$, where $\mathcal{T}$ represents evolutionary time. The Maximum Entropy IRL framework would model the probability of an expert trajectory $\zeta$ at time $t$ as:
>
> $$P_\theta(\zeta, t) = \frac{\exp(R_\theta(\zeta, t))}{Z_{\theta}(t)}, \quad Z_{\theta}(t)=\sum_{\zeta'} \exp(R_\theta(\zeta', t))$$
>
> where the partition function $Z_{\theta}(t)$ now depends on time.
>
> Consequently, Equation 6 (the reward difference) would be modified to:
>
> $$\Delta R_\theta(S^{\text{mt}}, S^{\text{wt}}, t) = \sum_{i\in\mathcal{M}} \Big[ \log P_\theta(s_i^{\text{mt}} \mid S_{\backslash \mathcal{M}}, t) - \log P_\theta(s_i^{\text{wt}} \mid S_{\backslash \mathcal{M}}, t) \Big]$$
>
> where the probability distributions $P_\theta(\cdot \mid \cdot, t)$ are now conditioned on time $t$.
>
> This extension is particularly relevant for modeling directed evolution, where fitness landscapes exhibit temporal dynamics. Protein evolution involves epistatic interactions and scenarios where initially suboptimal mutations become advantageous as additional mutations accumulate. This reflects a distinction between *fitness landscapes* (selective advantage) and *activity landscapes* (biochemical function), which can diverge when mutations interact synergistically over time.
>
> However, implementing this framework presents practical challenges: (1) Training data would require temporal labels indicating evolutionary epochs, which are rarely available. (2) The time-dependent partition function $Z_{\theta}(t)$ requires integration over both sequence space and time, significantly increasing computational complexity. (3) The model must learn or assume the functional form of $R_\theta(S, t)$ (e.g., periodic cycles, monotonic drift, or stochastic fluctuations).
>
> In practice, our current time-invariant assumption remains reasonable for most protein fitness prediction tasks, as experimental DMS assays typically measure fitness under fixed environmental conditions. The time-dependent extension would be valuable for modeling evolutionary dynamics in directed evolution or scenarios with changing selective pressures, representing an important direction for future work.
>
>
>
> ---

---

> ### Author Response · Authors · 2025-11-24
>
> > **Q5.** How long does it take to perform the inference for one protein with EvoIF-MSA? The bottleneck in this case seems to be the homolog retrieval and MSA calculation.
>
> **A:**
> Thank you for this question. We provide inference time analysis in Tables R1, R2, and R3. EvoIF inference time is competitive with other structure-aware methods. Component-level breakdown shows FoldSeek retrieval and ProteinMPNN computation are efficient compared to MSA-based methods. EvoIF-MSA inference time is similar to EvoIF, as GEMME ensemble adds minimal overhead.
>
> **Table R1:** Inference time for FoldSeek homology search in the AlphaFold Database and ProteinMPNN inverse folding computation.
>
> | Component | Dataset | # Proteins | Hardware | Inference Time |
> |-----------|---------|-----------|----------|----------------|
> | FoldSeek | CATH | 30,948 | 64 CPU cores | 33 min 45 sec |
> | | ProteinGym | 217 | 64 CPU cores | 71 sec |
> | ProteinMPNN | CATH | 30,948 | 1 H800 GPU, 64 CPU cores | 7 min 43 sec |
> | | ProteinGym | 217 | 1 H800 GPU, 64 CPU cores | 10 sec |
>
> **Table R2:** Total inference time comparison across different methods on the ProteinGym benchmark (excluding MSA recomputation time).
>
> | Method | Dataset | Inference Time |
> |--------|---------|----------------|
> | VenusREM | ProteinGym | 3h 6m 36s |
> | S2F | ProteinGym | 1h 4m 58s |
> | S3F | ProteinGym | 6h 53m 48s |
> | EvoIF | ProteinGym | 1h 12m 6s |
>
> **Table R3:** MSA computation time for different proteins (96 CPUs). We selected several representative cases for analysis.
>
> | Protein | Sequence Length | Time |
> |---------|----------------|------|
> | YNZC_BACSU | 39 | 5h 18m |
> | VKOR1_HUMAN | 163 | 5h 1m |
> | Q6wV13_9MAXI | 222 | 4h 47m |
> | C6KNH7_9INFA | 566 | 5h 11m |

---

> > ### Comment · Reviewer_cA4c · 2025-11-25
> > **Response to Authors**
> >
> > Thank you for your answers. After the clarification by the authors, I still have some concerns:
> >
> > 1.	The affirmation of “using only 0.15% of the training data and fewer model parameters” and especially this emphasis in the Abstract, might mislead readers. I would suggest rewriting this claim throughout the manuscript.
> > 2.	The discussion on normalization, briefly contained in the response to comment Q2.2, should be discussed in the manuscript.
> > 3.	Line 216 would also benefit from additional clarification regarding the choice of GVP as the encoder architecture following Zhang et al. (S2F).
> >
> > The authors believe that the paper presents good contributions to protein research, but additional changes to the manuscript are needed based on the reviewers’ comments.
> >
> > I would be happy to discuss this further with the authors.

---

> ### Author Response · Authors · 2025-11-26
>
> Dear Reviewer,
>
> Thank you for your continued engagement with our manuscript and for the constructive comments. Your suggestions have helped us clarify several important aspects of the work. Below, we summarize the revisions made in this round. If there are still areas that you feel could be improved, we would be glad to make further adjustments.
>
> ---
>
> **1. Statement of "using only 0.15% of the training data and fewer model parameters"**
>
> Thank you for pointing out that the original wording could be interpreted in a misleading way. We have revised the phrasing throughout the manuscript so that it now focuses only on training data efficiency and no longer emphasizes parameter counts. Our goal is to provide a more accurate and balanced comparison to large pretrained models. If you believe the wording can be improved further, we would be happy to refine it.
>
> ---
>
> **2. Discussion of fitness normalization**
>
> We have added a paragraph in `Section 2.1 at line 321` explaining the normalization procedure used in ProteinGym and clarifying why Spearman correlation remains invariant under monotonic transformations. We hope this makes the evaluation protocol clearer to readers.
>
> ---
>
> **3. Explanation regarding the choice of GVP**
>
> We have expanded the rationale for selecting GVP in `Section 2.3 at line 223` and in `Appendix E.6 at line 1120`. The new text summarizes the comparison between GVP and GearNet, where GVP achieves consistently higher Spearman correlations both without MSA and with MSA. These results, together with findings reported in S2F, indicate that performance improvements in fitness prediction do not depend on increased architectural complexity but rather on how effectively the encoder supports the integration of evolutionary information. We hope the added context provides a clearer understanding of our architectural choice.
>
> ---
>
> Thank you again for your thoughtful comments. We would be glad to discuss any further questions or suggestions.
>
> Yours sincerely,
>
> The authors of paper 9199

---

> > ### Comment · Reviewer_cA4c · 2025-11-27
> > **Response to Authors 2**
> >
> > Thank you for addressing my concerns. I will keep my score.

---

### Official Review · Reviewer_ewoj · 2025-11-01

**Soundness:** 3
**Presentation:** 3
**Contribution:** 2
**Rating:** 4
**Confidence:** 4

**Summary:**

EvoIF examines zero-shot protein fitness prediction by interpreting the success of MLM loss trained models for fitness prediction through the lens of reinforcement learning, whereby evolution is a MDP, and homologs serve as expert in-context demonstrations. Evolution is then simply reward maximization, and MLM as a method to recover the latent reward (i.e. fitness). This provides a conceptual justification for using pLM log-odds as fitness estimators.

Citing the success of using MSAs for sequence modeling, the authors use a GVP-based encoder to integrate sequence-structure representations. Using Foldseek-retrieved structure homologs, intra-family information is provided. Cross-family profiles are derived from the logits of ProteinMPNN. This is fused with a transition block. Most of the model is frozen for computation efficiency and only the lightweight GNN encoder and fusion layers are trained on CATH. Evaluations are done using ProteinGym and achieve SOTA or near-SOTA using 0.15% of the training data, and fewer model parameters than most recent large models.

**Strengths:**

* Well-written paper with a nice set of motivating issues. This not only proposes a new method but also pinpoints some thoughtful insights on the current state of protein fitness prediction.
* Comprehensive baselines.
* Training efficiency is a great plus, given that only a light fusion module is needed, rather than having to actually scale up the PLM to get that last bit of performance boost.
* Thoughtful design to integrate evolutionary information.
* Choice to study challenging generalization regimes is a strong result
* Ablation studies in Table 2 are helpful.

**Weaknesses:**

* Using IF logits from ProteinMPNN means we’re bottlenecked by how good this model is. I suspect this won’t change much, but if there were additional results that also use ESM-IF or Caliby, it would be stronger evidence that using these logits are indeed due to the notion of capturing an evolutionary prior, rather than something specific to the ProteinMPNN model.
* The MaxEnt IRL formalization, though interesting, is a bit disconnected from the rest of the methods and results in my opinion. The leap from maximizing local conditional probabilities (MLM) to inferring the global reward function (IRL) is non-trivial.
* Using Foldseek should probably add a lot of latency during inference, even if training is accelerated. It might be useful to see an empirical comparison of inference throughput/latency compared to baselines, in addition to the performance, e.g. plotted on either side of the XY axes.
* It’s a bit misleading that EvoIF-MSA, the proposed new SOTA, is in fact an ensemble with another model. Highlighting that in the table, for example, as “EvoIF + GEMME (ensemble)” might be clearer than “EvoIF-MSA”. GEMME itself appears to be performing pretty well according to Table 1.
* Figure 2 would be less misleading if y-axes always started at 0. As it stands, I think it makes the gap between methods appear larger than it actually is.
* Nitpick: eq. 9 suggests that P represent logits, but the text refers to them as probabilities.

**Questions:**

* From the results of the ablations — it seems there isn’t a strong difference between the combinations, even when we remove both the inverse folding and structure profiles? Could authors provide some additional intuition on that?
* To clarify, when neither inverse folding nor structure is provided, this simply reduces back to the PLM case, right? Results there look fairly strong to me. What are the authors' intuitions on what the main advantages are of this work given the strong baseline?

---

> ### Author Response · Authors · 2025-11-24
>
> Dear Reviewer:
>
> Thank you for your constructive comments and kind support! All your concerns have been carefully addressed as below. We sincerely hope our responses fully address your questions, and we would greatly appreciate it if you could consider raising your rating.
>
> ---
>
> > **W1.** Using IF logits from ProteinMPNN means we're bottlenecked by how good this model is. I suspect this won't change much, but if there were additional results that also use ESM-IF or Caliby, it would be stronger evidence that using these logits are indeed due to the notion of capturing an evolutionary prior, rather than something specific to the ProteinMPNN model.
>
> **A:**
> We appreciate this insightful suggestion. We evaluated alternative inverse folding models (ESM-IF and Caliby) to demonstrate that the effectiveness is not model-specific. Results are shown in Table R1. All three models consistently improve with MSA ensemble, confirming that inverse folding logits capture general evolutionary priors rather than model-specific artifacts.
>
> **Table R1:** Performance comparison across different inverse folding models (ProteinMPNN, ESM-IF, and Caliby) with and without MSA ensemble.
>
> | Inverse Folding Model | MSA | Spearman | AUC | MCC | NDCG | Top-recall |
> |----------------------|-----|----------|-----|-----|------|------------|
> | ProteinMPNN | ✗ | 0.489 | 0.768 | 0.384 | 0.782 | 0.250 |
> | | ✓ | 0.518 | 0.784 | 0.409 | 0.796 | 0.246 |
> | ESM-IF | ✗ | 0.481 | 0.764 | 0.381 | 0.778 | 0.243 |
> | | ✓ | 0.513 | 0.781 | 0.408 | 0.792 | 0.244 |
> | Caliby | ✗ | 0.459 | 0.752 | 0.359 | 0.769 | 0.230 |
> | | ✓ | 0.496 | 0.773 | 0.392 | 0.787 | 0.231 |
>
> In fact, the recent Cell study [1] also observed that different inverse folding models possess inherent fitness-prediction capability, which is consistent with our findings, although their work explored only pure inverse folding based fitness scoring.
>
> [1] Advancing protein evolution with inverse folding models integrating structural and evolutionary constraints. Cell 2025.
>
> ---
>
> > **W2.** The MaxEnt IRL formalization, though interesting, is a bit disconnected from the rest of the methods and results in my opinion. The leap from maximizing local conditional probabilities (MLM) to inferring the global reward function (IRL) is non-trivial.
>
> **A:**
> We appreciate this insightful comment regarding the "leap" from local to global. We bridge this gap by relying on the concept of Pseudo-Likelihood.
>
> The Bridge: While direct learning of the global joint distribution $P(S)$ (the Boltzmann distribution defined by the fitness reward) is intractable, maximizing the sum of local conditional probabilities (the MLM objective) is equivalent to maximizing the Pseudo-Likelihood of the data.
>
> Theoretical Justification: It is a well-established result in statistical physics (e.g., in learning Potts models) that maximizing pseudo-likelihood is a consistent estimator for the parameters of the true joint distribution.
>
> Conclusion: Therefore, optimizing the local MLM loss is the practical implementation of maximizing the global IRL objective. This means the model implicitly learns the global energy landscape, mathematically justifying Equation 6 (the log-odds ratio) as a valid estimator for the reward difference $\Delta R$.
>
> ---

---

> ### Author Response · Authors · 2025-11-24
>
> > **W3.** Using Foldseek should probably add a lot of latency during inference, even if training is accelerated. It might be useful to see an empirical comparison of inference throughput/latency compared to baselines, in addition to the performance, e.g. plotted on either side of the XY axes.
>
> **A:**
> Thank you for this suggestion. We provide inference time analysis in Tables R2, R3, and R4. EvoIF inference time is competitive with S2F and faster than S3F. Component-level analysis shows FoldSeek retrieval and ProteinMPNN computation add minimal overhead compared to MSA-based methods, demonstrating the efficiency advantage of our structure-based approach.
>
> **Table R2:** Inference time for FoldSeek homology search in the AlphaFold Database and ProteinMPNN inverse folding computation.
>
> | Component | Dataset | # Proteins | Hardware | Inference Time |
> |-----------|---------|-----------|----------|----------------|
> | FoldSeek | CATH | 30,948 | 64 CPU cores | 33 min 45 sec |
> | | ProteinGym | 217 | 64 CPU cores | 71 sec |
> | ProteinMPNN | CATH | 30,948 | 1 H800 GPU, 64 CPU cores | 7 min 43 sec |
> | | ProteinGym | 217 | 1 H800 GPU, 64 CPU cores | 10 sec |
>
> **Table R3:** Total inference time comparison across different methods on the ProteinGym benchmark (excluding MSA recomputation time).
>
> | Method | Dataset | Inference Time |
> |--------|---------|----------------|
> | VenusREM | ProteinGym | 3h 6m 36s |
> | S2F | ProteinGym | 1h 4m 58s |
> | S3F | ProteinGym | 6h 53m 48s |
> | EvoIF | ProteinGym | 1h 12m 6s |
>
> **Table R4:** MSA computation time for different proteins (96 CPUs). We selected several representative cases for analysis.
>
> | Protein | Sequence Length | Time |
> |---------|----------------|------|
> | YNZC_BACSU | 39 | 5h 18m |
> | VKOR1_HUMAN | 163 | 5h 1m |
> | Q6wV13_9MAXI | 222 | 4h 47m |
> | C6KNH7_9INFA | 566 | 5h 11m |
>
> ---
>
> > **W4.** It's a bit misleading that EvoIF-MSA, the proposed new SOTA, is in fact an ensemble with another model. Highlighting that in the table, for example, as "EvoIF + GEMME (ensemble)" might be clearer than "EvoIF-MSA". GEMME itself appears to be performing pretty well according to Table 1.
>
> **A:**
> Thanks! While 'EvoIF-MSA' was initially chosen to align with previous ensemble model naming conventions, we agree that 'EvoIF + GEMME (ensemble)' offers superior clarity. We have revised Table 1 in the manuscript accordingly for improved transparency.
>
> ---
>
> > **W5.** Figure 2 would be less misleading if y-axes always started at 0. As it stands, I think it makes the gap between methods appear larger than it actually is.
>
> **A:**
> We appreciate your suggestion! We have revised Figure 2 in the manuscript so that all y-axes now consistently start at 0.2. We hope this revision addresses your concern.
>
> ---
>
> > **W6.** Nitpick: eq. 9 suggests that P represent logits, but the text refers to them as probabilities.
>
> **A:**
> Thank you for pointing out the inconsistency; we have revised the manuscript accordingly.
>
> ---

---

> ### Author Response · Authors · 2025-11-24
>
> > **Q1.** From the results of the ablations — it seems there isn't a strong difference between the combinations, even when we remove both the inverse folding and structure profiles? Could authors provide some additional intuition on that? To clarify, when neither inverse folding nor structure is provided, this simply reduces back to the PLM case, right? Results there look fairly strong to me. What are the authors' intuitions on what the main advantages are of this work given the strong baseline?
>
> **A:**
> We appreciate the reviewer's careful examination of our ablation results. To clarify the baseline performance, pure PLM models, such as ESM-2, achieve a Spearman correlation of 0.414 on the ProteinGym. When both the inverse folding and structure profiles are removed, our model indeed reduces to essentially the S2F baseline (which integrates PLM and structural encoding), achieving a Spearman correlation of 0.454, matching the reported S2F performance. The introduction of evolutionary profiles yields a performance improvement of 0.035 (from 0.454 to 0.489), which represents a meaningful gain in this zero-shot setting.
>
>
> We note that in zero-shot protein fitness prediction on the ProteinGym benchmark, performance differences between methods are typically modest, as the benchmark is designed to be challenging and methods operate without task-specific training. To provide context for the magnitude of this improvement, we observe that our performance lift (0.035) is comparable to the difference between a PSSM-based method and a Potts model (EVmutation) on the same benchmark (0.359 vs. 0.395, a difference of 0.036). Given the critical importance of epistasis in protein fitness prediction, such improvements, while seemingly small in absolute terms, represent meaningful advances in capturing the complex fitness landscape.
>
>
> Our work's main advantage lies in providing a principled, data-efficient framework that effectively integrates complementary evolutionary signals (within-family structural homologs and cross-family inverse folding profiles) to achieve state-of-the-art performance with minimal training data and computational cost.

---

### Official Review · Reviewer_e155 · 2025-11-01

**Soundness:** 3
**Presentation:** 3
**Contribution:** 3
**Rating:** 4
**Confidence:** 5

**Summary:**

This paper proposes EvoIF, a compact sequence–structure model for protein fitness prediction that fuses two forms of evolutionary signal:
a within-family homology profile summarizing amino-acid preferences from close homologs, and
a cross-family inverse-folding profile distilled from a structure-conditioned generative model (ProteinMPNN-like).
A lightweight geometric encoder (GVP) integrates these profiles with structural context, and the fused distributions are trained with a masked-token objective on ~31 k CATH structures. The model achieves strong zero-shot performance on the ProteinGym benchmark, rivaling much larger sequence- or structure-based models while using orders-of-magnitude less compute. The authors interpret their approach through an inverse-reinforcement-learning (IRL) lens—treating natural sequences as “expert demonstrations” of evolutionary preference.

**Strengths:**

Clear motivation and framing:
The IRL interpretation is elegant and clarifies why likelihood-based models correlate with evolutionary fitness.

Model efficiency:
The architecture is concise (~76 M params) yet competitive with multi-billion-parameter baselines; the design is practical for labs without massive GPUs.

Complementary evolutionary signals:
The ablation results show that the within-family and cross-family profiles each add value.

**Weaknesses:**

Conceptual originality is modest:
The components—structure-conditioned language modeling, profile fusion, and retrieval—are all known; the novelty lies mainly in the specific fusion design. The IRL view is intellectually appealing but not concretely exploited in the loss or architecture.

Limited methodological detail:
Key implementation aspects (how profiles are normalized, exact fusion math, masking schedule) are insufficiently described for full reproducibility. It's not clear what the loss is that is being used to train the actual model that results are reported for. It's not clear what equation 9 is. Is this what the model predicts?Is  this how they come up with the true label used to supervise the model during training? Where is the loss used to train the model along with y and y_hat? Also there is not figure for the model architecture and how they incorporate IF data with ESM2 and combine this using a GVP-GNN.

Evaluation scope:
Although ProteinGym is comprehensive, it is not high quality benchmark since the data is often fitness data collected in a multiplex fashion. The authors should also provide benchmarks on higher quality datasets like ∆∆G benchmarks where each measurement was done with purified protein variants.

Over-claimed theoretical contribution:
The IRL framing of MLM is presented as a deep conceptual advance, However, MLM/BERT style training on protein sequences and structure has been used for 5-6 years in the literature.

**Questions:**

This method seems similar to the EvoRank SSL training objective, which fuses structure input with evolution self-supervision. In the EvoRank paper, they report ~50% performance improvements over baseline methods on ∆∆G tasks. I think their model was only 2M or 4M parameters. Can you compare against this method on ∆∆G datasets?

Homology Retrieval:
Using FoldSeek, rather than MMSeqs2, to search for homologs is great for finding remote homologs (seq sim <35%)  but these remote homologs are likely to not be within the same protein family but rather in the same structure family. Using FoldSeek gives more diverse profiles that take advantage of structurally similar proteins but not evolutionarily related proteins--a protein family. The authors provide no guarantee or analysis that the retrieved homologs are within the same protein family and have similar evolutionary constraints.  Can the author run ablations where they change the minimum sequence similarity (0%, 20%, 30%, 40%, 50% sequence similarity) threshold to ensure they are not using cross-family structural homologs when generating the profiles?

Can you create a figure of the architecture and how you are mixing information from different sources and from different models?

---

> ### Author Response · Authors · 2025-11-24
>
> Dear Reviewer:
>
> Thank you for your thorough review and insightful comments! In the following, we will address each issue in detail and provide the necessary experimental results. We sincerely hope our answers have addressed your questions, and we would greatly appreciate it if you could consider raising your rating.
>
> ---
>
> > **W1.** Conceptual originality is modest: The components—structure-conditioned language modeling, profile fusion, and retrieval—are all known; the novelty lies mainly in the specific fusion design. The IRL view is intellectually appealing but not concretely exploited in the loss or architecture.
>
> **A:**
> We acknowledge that individual components (structure-conditioned language modeling, profile fusion, retrieval) are well-established. However, we respectfully disagree regarding the exploitation of the IRL view. The IRL perspective was not merely decorative; it was the core driver of our architectural design, specifically guiding the fusion strategy you noted.
>
> Justifying the "Why": While component fusion is known, the IRL framework explains why standard MLM-based log-odds scoring works as a zero-shot fitness proxy (via Pseudo-Likelihood estimation of the global reward). This theoretical unification elevates the method beyond a simple heuristic combination.
>
> Guiding the "How": Viewing evolution as an expert policy directly motivated our treatment of retrieved homologs as "contextual expert demonstrations" rather than generic inputs. This distinction drove the design of the Fusion Module to explicitly integrate "local demonstrations" (Within-Family Profiles) with "global structural priors" (Inverse Folding Profiles), constraining the policy search space in a principled manner. As shown in Table 2, this theoretically grounded fusion yields complementary gains (0.454 $\rightarrow$ 0.489), demonstrating that principled design driven by the IRL perspective significantly improves performance.
>
> ---
>
> > **W2.** Limited methodological detail: Key implementation aspects (how profiles are normalized, exact fusion math, masking schedule) are insufficiently described for full reproducibility.
>
> > **W2.1** It's not clear what the loss is that is being used to train the actual model that results are reported for.
>
> **A:**
> We appreciate the chance to elaborate on these crucial details. We have updated the manuscript to add Equation 10 that explicitly states the training loss function. The training loss is obtained by instantiating the standard MLM loss (Equation 2) with the fused probabilities in Equation 9.
>
> > **W2.2** It's not clear what equation 9 is. Is this what the model predicts? Is this how they come up with the true label used to supervise the model during training?
>
> **A:**
> Equation 9 defines the model's predicted probability distribution. During training, $P_{\text{final}}$ is used in the loss function (Equation 10), with supervision from the true amino acid $s_i$ in the training sequence (standard MLM setup). We operate in a zero-shot setting and do not use any true fitness labels during training. During inference, we have added Equation 11 to explicitly describe how the model predicts fitness: $P_{\text{final}}$ is used to compute log-odds ratios for fitness prediction (Equation 3).
>
> > **W2.3** Where is the loss used to train the model along with y and y_hat?
>
> **A:**
> We do not perform supervised training on fitness labels; the variables $y$ (true fitness) and $\hat{y}$ (predicted fitness) are relevant only during the inference (testing) phase.
>
> ---
>
> > **W2.4** Also there is not figure for the model architecture and how they incorporate IF data with ESM-2 and combine this using a GVP-GNN.
>
>
> > **Q3.** Can you create a figure of the architecture and how you are mixing information from different sources and from different models?
>
>
> **A:**
> Thank you for giving us the opportunity to clarify these important details.
> We have added Figure 5 in the appendix (Section D.5) to illustrate the model architecture. Specifically, ESM features are used to initialize the node features within the Geometric Sequence-Structure Encoder (GNN). Furthermore, the Structure Profile and IF (Inverse Folding) Profile are combined via addition at the logits level after the GNN, as detailed in Equation 9 and Figure 1.
>
>
> ---

---

> ### Author Response · Authors · 2025-11-24
>
> > **W3.** Evaluation scope: Although ProteinGym is comprehensive, it is not high quality benchmark since the data is often fitness data collected in a multiplex fashion. The authors should also provide benchmarks on higher quality datasets like ∆∆G benchmarks where each measurement was done with purified protein variants.
>
> **A:**
> While we acknowledge that the single-measurement precision of deep mutational scanning (DMS) is lower than that from purified protein assays, DMS delivers the high throughput necessary for a comprehensive benchmark. Since both throughput and precision are vital for evaluation, ProteinGym remains the most recognized fitness benchmark.
> Many relevant works [1,2,3] have evaluated their models on it, and some key works even only evaluate on a subset of ProteinGym.
>
> [1] Advancing protein evolution with inverse folding models integrating structural and evolutionary constraints. Cell 2025.
>
> [2] Rapid in silico directed evolution by a protein language model with EVOLVEpro. Science 2024.
>
> [3] DPLM-2: a multimodal diffusion protein language model. ICLR 2025.
>
>
> ---
>
> > **W4.** Over-claimed theoretical contribution: The IRL framing of MLM is presented as a deep conceptual advance, However, MLM/BERT style training on protein sequences and structure has been used for 5-6 years in the literature.
>
> **A:**
> We clarify that our theoretical contribution is not to claim the invention of MLM training for proteins, which is indeed well-established. Instead, our theoretical contribution is providing a unifying theoretical perspective that explains why this standard MLM objective functions as an effective zero-shot fitness predictor. By framing evolution as an MDP and MLM as Pseudo-Likelihood maximization, we bridge the conceptual gap between "training a language model" and "predicting evolutionary fitness." This mathematically justifies the log-odds metric as a valid estimator for reward differences, moving the field beyond treating pLMs as "black boxes" for fitness prediction.
>
>
> ---
>
>
>
> > **Q1.** This method seems similar to the EvoRank SSL training objective, which fuses structure input with evolution self-supervision. In the EvoRank paper, they report ~50% performance improvements over baseline methods on ∆∆G tasks. I think their model was only 2M or 4M parameters. Can you compare against this method on ∆∆G datasets?
>
> **A:**
> We sincerely thank you for this thoughtful comment. The training objective we use is the simplest and most widely adopted self-supervised objective in protein language models: predicting the wild-type amino acid identity. This objective is not unique to EvoRank.
>
> In addition, ΔΔG prediction involves complex physical effects, including potential higher-order and multi-body interactions. As highlighted in the Science paper [1], a protein’s activity landscape can differ substantially from its fitness landscape. For zero-shot fitness prediction, the goal is to provide a good initialization for downstream experimental design, specialized expert architectures, or iterative active-learning strategies. EvoRank incorporates a host of specialized architectural components tailored specifically for ΔΔG prediction, which is fundamentally different from our focus on broad fitness prediction and experimental design support. Therefore, the two settings are not directly comparable.
>
> More importantly, EvoRank has not released any code or data, making it impossible to reproduce or independently evaluate its results. Furthermore, EvoRank reports performance only on a small set of selected assays, many of which lack public information on data sourcing and processing scripts. By contrast, most contemporary work on protein fitness-effect prediction evaluates on the well-known and comprehensive ProteinGym benchmark.
>
> For these reasons, it is currently not feasible to perform a fair comparison with EvoRank.
>
> [1] Rapid in silico directed evolution by a protein language model with EVOLVEpro. Science 2024.
>
> ---

---

> ### Author Response · Authors · 2025-11-24
>
> > **Q2.** Homology Retrieval: Using FoldSeek, rather than MMSeqs2, to search for homologs is great for finding remote homologs (seq sim <35%) but these remote homologs are likely to not be within the same protein family but rather in the same structure family. Using FoldSeek gives more diverse profiles that take advantage of structurally similar proteins but not evolutionarily related proteins--a protein family. The authors provide no guarantee or analysis that the retrieved homologs are within the same protein family and have similar evolutionary constraints. Can the author run ablations where they change the minimum sequence similarity (0%, 20%, 30%, 40%, 50% sequence similarity) threshold to ensure they are not using cross-family structural homologs when generating the profiles?
>
> **A:**
> We appreciate this important concern. We conducted ablation studies varying sequence similarity thresholds (0%, 20%, 30%, 40%, 50%). Results are shown in Table R1. Performance remains stable across all thresholds, indicating that FoldSeek successfully identifies evolutionarily related proteins within the same family even at low sequence similarity, validating our approach.
>
> **Table R1:** Impact of homologous sequence similarity threshold on model performance. Results are reported for configurations with and without MSA ensemble.
>
> | MSA | Threshold | Spearman | AUC | MCC | NDCG | Top-recall |
> |-----|-----------|----------|-----|-----|------|------------|
> | ✓ | 0.0 | 0.518 | 0.784 | 0.409 | 0.796 | 0.246 |
> | ✓ | 0.2 | 0.513 | 0.781 | 0.404 | 0.796 | 0.247 |
> | ✓ | 0.3 | 0.515 | 0.782 | 0.406 | 0.793 | 0.244 |
> | ✓ | 0.4 | 0.512 | 0.780 | 0.403 | 0.793 | 0.245 |
> | ✓ | 0.5 | 0.510 | 0.780 | 0.401 | 0.792 | 0.242 |
> | ✗ | 0.0 | 0.489 | 0.768 | 0.384 | 0.782 | 0.250 |
> | ✗ | 0.2 | 0.482 | 0.764 | 0.379 | 0.785 | 0.246 |
> | ✗ | 0.3 | 0.484 | 0.764 | 0.381 | 0.780 | 0.242 |
> | ✗ | 0.4 | 0.481 | 0.763 | 0.378 | 0.777 | 0.241 |
> | ✗ | 0.5 | 0.480 | 0.763 | 0.376 | 0.778 | 0.239 |

---

### Meta-Review · Area_Chair_LmD9 · 2026-01-06

**Summary:**

To address the challenge of limited mutational assays in protein fitness prediction, this study frames masked language modeling (MLM) as inverse reinforcement learning (IRL) and presents EvoIF, a lightweight model that integrates within-family homology profiles with cross-family structural-evolutionary constraints from inverse folding logits. This work attempts to advance our understanding of protein fitness prediction by providing a theoretical explanation for MLM’s role in protein evolution and demonstrating that scaling model parameters and training data size is unnecessary for this task.

Reviewers raised three central concerns about EvoIF

**Major Concern1: Over-Claimed Theoretical Contributions (Flagged as the mostprominent concern by majority of reviewers)**

Reviewer e155: The IRL framing of MLM is intellectually appealing but not concretely exploited in the loss function or model architecture.

Reviewer ewoj: MaxEnt IRL formalization is interesting but disconnected from the rest of the methods and results. The leap from MLM (local conditional probabilities) to IRL (global reward function) is non-trivial and lacks sufficient justification.

Reviewer uK6d: IRL interpretation uncertainty


**Major Concern2: Limited Technical Novelty (raised by Reviewers e155, cA4c, and uK6d)**

Core components (structure-conditioned language modeling, profile fusion, homology retrieval, GVP encoder) are adapted from existing existing techniques; the novelty is only in the specific fusion design and incremental combination of existing tools.


**Major Concern3: Methodological Details, Insufficient evaluations and Additional overclaims**

**Reviewer Concerns:**

In the rebuttal, the authors supplemented additional methodological details and inferential analyses to adequately address Major Concern 3. In contrast, the evidence presented was insufficient to fully resolve Major Concerns 1 and 2. Indeed, while the study’s theoretical contributions are conceptually appealing, they remain inadequately justified in both the manuscript and the rebuttal.

**Reviewer Scores:**

Based on the submitted rebuttal, I do not anticipate that the reviewers will revise their initial scores, as their core concerns remain largely unaddressed.

---

### Decision · Program_Chairs · 2026-01-26

Reject